# SparseOpt: Addressing Normalization-induced Gradient Skew in Sparse Training

Mohammed Adnan [1 2]   Rohan Jain [1]   Tom Jacobs [3]   Ekansh Sharma [4 2]   Rahul G. Krishnan [4 2]

Rebekka Burkholz [3]   Yani Ioannou [1]

## Abstract

Dynamic Sparse Training (DST) methods train neural networks by maintaining sparsity while dynamically adapting the network topology. Despite the promise of reduced computation, DST methods converge significantly slower than dense training, often requiring comparable training time to achieve similar accuracy. We demonstrate both analytically and empirically that Batch Normalization (BN) adversely affects sparse training, and propose **SparseOpt** — a sparsity-aware optimizer — to address this. Experiments on ResNet models across CIFAR-100 and ImageNet demonstrate consistently faster convergence and improved generalization with our proposed method. Our work highlights the limitations of current normalization layers in sparse training and provides the first systematic study of the interaction between Batch Normalization, sparse layers, and DST, taking a significant step toward making DST practically competitive with dense training.

## 1. Introduction

In recent years, Deep Neural Networks (DNNs) have achieved remarkable performance across a diverse range of tasks. This success can largely be attributed to the exponential increase in model size (Kaplan et al., 2020), and corresponding increase in the cost of training and deploying large models. To address this, model pruning and quantization have gained significant traction as approaches to reduce computational requirements for deploying foundational models. In contrast to post-training model compression methods, sparse training can leverage hardware-amenable weight sparsity during training, including structured sparsity (Lasby et al., 2024; Tyagi et al., 2025), making it potentially more computationally efficient. Sparse training has also found applications outside efficiency, including in adversarial robustness (Wu et al., 2025) and recently mechanistic interpretability in Large Language Models (LLMs) (Gao et al., 2025).

Dynamic Sparse Training (DST) is widely considered the state-of-the-art sparse training approach; DST methods continuously adapt the sparse topology (mask) during training and can match dense model generalization while maintaining high sparsity throughout training (Mocanu et al., 2018; Evci et al., 2020). Despite the promise of DST in enabling training efficiency, in practice DST exhibits much slower convergence as compared to dense training, requiring as many as five times more epochs to reach the same generalization performance (Evci et al., 2020). As a result, existing DST methods offer little to no reduction in total training time in practice compared to dense models, even considering methods benefiting from hardware acceleration. For sparse training to be practically useful, it is therefore essential to improve the convergence rate of sparse training methods such as DST.

Batch Normalization (BN), proposed by Ioffe & Szegedy (2015), was introduced to improve training stability and convergence rates in DNNs. While several works have analyzed BN in the context of dense models, the role of normalization layers in Sparse Neural Networks (SNNs) has not been studied. In this work, we aim to understand the impact of BN on the training dynamics and convergence rate of SNNs. We theoretically and empirically show that BN introduces training instability by altering the direction and magnitude of gradients in SNNs, depending on neuron sparsity. In the specific context of DST, this in turn leads to sudden changes in gradient direction during the mask update step, resulting in exacerbated training instability and slower convergence. To mitigate this issue, we propose a preconditioned gradient descent algorithm for sparse training that preserves the gradient direction during the mask update step by accounting for the varying sparsities of different neurons (i.e. differing fan-in across neurons).

In this work, we study the interaction between normalization layers and sparse training dynamics, and make the following key contributions:

1. A theoretical analysis on the impact of Batch Normalization on the training dynamics of Sparse Neural Networks, specifically we show that for DST, BN can

---

[1]University of Calgary [2]Vector Institute [3]CISPA Helmholtz Center for Information Security [4]University of Toronto. Correspondence to: Mohammed Adnan <adnan.ahmad@ucalgary.ca>.

*Proceedings of the 43rd International Conference on Machine Learning*, Seoul, South Korea. PMLR 306, 2026. Copyright 2026 by the author(s).

change the gradient direction during training.

2. We propose a novel preconditioned gradient descent optimization method for sparse training, and provide novel insights on the interaction between BN and the state-of-the-art sparse training methodology: Dynamic Sparse Training (DST).

3. An empirical analysis of DST across multiple datasets and model architectures demonstrating improved convergence with our proposed method and better generalization within typical training schedules.

## 2. Background

**Training Dynamics.** Modern DNNs rely on the composition of affine transformations and non-linearities to model complex functions. Although this architecture allows DNNs to learn complex and expressive features, it simultaneously introduces optimization challenges such as vanishing or exploding gradients, which make training deep networks difficult as network depth increases (Glorot & Bengio, 2010; He et al., 2015; Ioffe & Szegedy, 2015).

**Initialization.** The initialization of parameters and the distributions of activations and gradients have been shown to be crucial for effective training, especially for very deep models. He et al. (2015) proposed a widely-used initialization for ReLU activation functions, based on that of Glorot & Bengio (2010) for sigmoids. Both are motivated by maintaining activation distributions of each layer with approximately unit variance, to mitigate the issue of exploding and vanishing gradients. Building on this, Evci et al. (2022) introduced a *sparsity-aware* initialization that accounts for heterogeneous incoming connections (fan-in) of neurons in a sparse layer.

**Normalization Layers.** To stabilize the training dynamics of deep networks, Ioffe & Szegedy (2015) introduced BN which has become an essential component of DNNs. Mechanistically, BN accelerates and stabilizes training by normalizing a layer's inputs through re-centering and re-scaling. Since the introduction of BN, there have been many other normalization layers proposed with similar motivation but different mechanisms (Ba et al., 2016; Wu & He, 2018).

**Preconditioned Gradient Descent.** Preconditioned gradient methods facilitate optimization by transforming the gradient vector via a preconditioning matrix $P_t$. In many popular optimizers, $P_t$ approximates the inverse Hessian or accumulated curvature statistics (Li, 2018), effectively rescaling the gradient to adapt to the local geometry of the loss landscape (Li et al., 2015). The goal of preconditioning is to accelerate convergence by mitigating the effects of pathological curvature and ill-conditioning in the optimization landscape (Martens & Grosse, 2015; Dauphin

et al., 2014). The general update rule for preconditioned gradient descent is defined as:

$$w^{t+1} = w^t - \eta P_t \nabla \mathcal{L}(w^t). \tag{1}$$

Stochastic Gradient Descent (SGD) is a special case of Equation (1), where the preconditioner $P_t$ is the identity matrix, $I$. Similarly, many common optimizers can be defined within this framework, including for example AdaGrad (Duchi et al., 2011) and RMSProp (Tieleman & Hinton, 2012). More recently, structured preconditioning methods like Shampoo (Gupta et al., 2018) have been proposed to capture dependencies between parameters via tensor products.

**Dynamic Sparse Training.** DST methods periodically update the sparse mask during training, effecting the exploration of a diverse set of topologies. Mocanu et al. (2018) proposed Sparse Evolutionary Training (SET), a DST method that at each mask update step prunes weights based on weight magnitude, and grows weights randomly. Rigging the Lottery Ticket (RigL), proposed by Evci et al. (2020), instead grows weights based on the dense gradient norm. While SET and RigL learn unstructured sparse masks, more recent work has shown DST can learn structured sparse masks amenable to hardware acceleration at inference (Lasby et al., 2024; Tyagi et al., 2025). In practice, DST methods are the state-of-the-art sparse training method, enabling generalization performance comparable to that of dense models, while maintaining high sparsity throughout training.

## 3. Revisiting Batch Norm. for Sparse Layers

In this section, we systematically study the effect of BN on sparse training. We analyze how heterogeneous connectivity, i.e. neurons with varying numbers of connections in a layer, alters the scale of the pre-activation variance and, consequently, the gradient induced by BN.

In the forward pass of a fully-connected neural network layer, for neuron $i$ and sample $b$, the pre-activation is:

$$x_i^{(b)} = \sum_{j=1}^{\mathcal{N}_{\ell-1}} w_{ij} h_j^{(b)}, \tag{2}$$

where $h_j^{(b)}$ denotes the activation of the $j$-th neuron in the previous layer for sample $b$, $w_{ij}$ is the weight connecting neuron $j$ to neuron $i$, and $\mathcal{N}_{\ell-1}$, which is the number of neurons in the previous layer.

Current BN implementations assume a fully-connected layer where $\text{fan}_{\text{in}}^i = \text{fan}_{\text{in}} \forall i$, i.e. every neuron has the same number of incoming connections. However, in the more general case of arbitrary connectivity on a layer, such as exhibited in sparse neural networks, $\text{fan}_{\text{in}}^i \leq \mathcal{N}_{\ell-1}$ *can vary for each neuron* $i$.

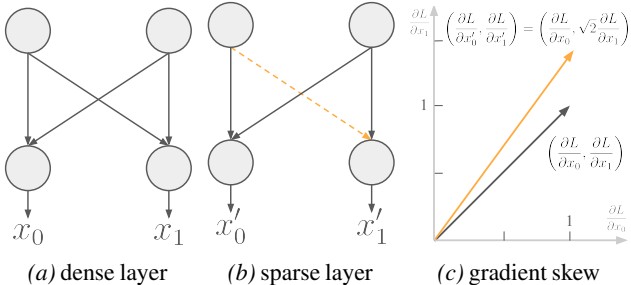

*(a) dense layer*    *(b) sparse layer*    *(c) gradient skew*

*Figure 1.* **Batch Normalization causes gradient skew in sparse layers**. BN scales gradients based on the variance for neurons in a dense layer (a). However, in a sparse layer with masked weights (dashed lines) (b), this can have the effect of scaling each gradient component differently. This non-uniform scaling effectively skews, i.e. rotates and scales, the gradient for a sparse layer (c). In (b), neuron $i = 1$ with pre-activation $x'_1$ has sparsity $s_1 = 0.5$. From Equation (11), $\partial L/\partial x'_1$ will be scaled by a factor of $1/\sqrt{1-s_i} = 1/\sqrt{0.5} = \sqrt{2}$ relative to $\partial L/\partial x_1$ in an equivalent dense layer (a).

In this more general case the pre-activation for neuron $i$ is:

$$x_i^{(b)} = \sum_{j=1}^{\mathrm{fan}_{\mathrm{in}}^i} w_{ij} h_j^{(b)}, \tag{3}$$

where $h_j^{(b)}$ denotes the activation of the $j$-th neuron in the previous layer for sample $b$, $w_{ij}$ is the weight connecting neuron $j$ to neuron $i$, and $\mathrm{fan}_{\mathrm{in}}^i$ is the number of incoming connections to neuron $i$.

Given a mini-batch of size $m$, BN computes the empirical mean and variance for each neuron:

$$\mu_i = \frac{1}{m} \sum_{b=1}^{m} x_i^{(b)}, \qquad \sigma_i^2 = \frac{1}{m} \sum_{b=1}^{m} (x_i^{(b)} - \mu_i)^2,$$

which are then used to normalize the pre-activations within the mini-batch, i.e.

$$\hat{x}_i^{(b)} = \frac{x_i^{(b)} - \mu_i}{\sigma_i}, \qquad y_i^{(b)} = \gamma_i \hat{x}_i^{(b)} + \beta_i.$$

Hence, the variance of the pre-activation, i.e. $\sigma_i^2$, determines the effective scale of both normalized activations and the gradients in the backward pass.

Consider the general form of the pre-activation $x_i^{(b)}$ as defined in Equation (3), assume $\{w_{ij}\}_j$ and $\{h_j^{(b)}\}_j$ are i.i.d., zero-mean random variables with $\mathrm{Var}[w_{ij}] = \sigma_w^2$ and $\mathrm{Var}[h_j^{(b)}] = \sigma_h^2$ respectively. Then each product $w_{ij} h_j^{(b)}$ has variance:

$$\mathrm{Var}[w_{ij} h_j^{(b)}] = \sigma_i^2 = \sigma_w^2 \sigma_h^2. \tag{4}$$

### 3.1. Layer with homogeneous connectivity (e.g. dense)

In a layer with homogeneous connectivity, every neuron has the same number of incoming connections, i.e.

$\forall i, \mathrm{fan}_{\mathrm{in}}^i = \mathrm{fan}_{\mathrm{in}}$. Note that dense layers are a special case of homogeneous connectivity where $\mathrm{fan}_{\mathrm{in}} = \mathcal{N}_{\ell-1}$, where $\mathcal{N}_{\ell-1}$ is the total number of neurons in the previous layer.

Since $x_i^{(b)}$ is a sum of $\mathrm{fan}_{\mathrm{in}}$ independent terms, the variance of $x_i^{(b)}$ is calculated as:

$$\mathrm{Var}[x_i^{(b)}] = \sum_{j=1}^{\mathrm{fan}_{\mathrm{in}}} \mathrm{Var}[w_{ij} h_j^{(b)}] = \mathrm{fan}_{\mathrm{in}} \sigma_w^2 \sigma_h^2. \tag{5}$$

Therefore, the corresponding standard deviation for the mini-batch depends on the total number of incoming connections:

$$\sigma_i = \sqrt{\mathrm{Var}[x_i^{(b)}]} = \sqrt{\mathrm{fan}_{\mathrm{in}}} \sigma_w \sigma_h, \tag{6}$$

showing that the BN normalization scale grows proportionally to the square root of the number of incoming connections.

### 3.2. Layer with heterogeneous connectivity (e.g. sparse)

In a layer with heterogeneous connectivity, every neuron $i$ may have different number of incoming connections $\mathrm{fan}_{\mathrm{in}}^i$. Sparse neural networks with unstructured sparse masks generally fall within this category of the more general case of a neural network layer.

The pre-activation $(x_i'^{(b)})$ for the heterogeneous layer can be then calculated as:

$$x_i'^{(b)} = \sum_{j=1}^{\mathrm{fan}_{\mathrm{in}}^i} w_{ij} h_j^{(b)},$$

As in Equation (5), and using Equation (4), the variance of the sparse activation can be similarly calculated as:

$$\mathrm{Var}[x_i'^{(b)}] = \mathrm{fan}_{\mathrm{in}}^i \sigma_w^2 \sigma_h^2 = \mathrm{fan}_{\mathrm{in}}^i \sigma_i^2, \tag{7}$$

Let $s_i = 1 - \mathrm{fan}_{\mathrm{in}}^i / \mathcal{N}_{\ell-1}$ denote the sparsity of neuron $i$:

$$\therefore \sigma_i' = \sqrt{1 - s_i} \sigma_i. \tag{8}$$

Thus sparsity reduces the normalization scale proportionally to $\sqrt{1 - s_i}$ or as sparsity $(s_i)$ increases variance decreases. For dense models, $s = 0$, i.e., all the neurons have the same number of incoming connection, and thus have same variance (assuming i.i.d.).

**Effect on gradients.** We next analyze how the pre-activation variance influences gradients $(\partial L/\partial x_i^{(b)})$ for each neurons through the backpropagation equations of Batch Normalization. The gradients can be derived as:

$$\frac{\partial L}{\partial x_i^{(b)}} = \frac{1}{\sigma_i} \left( \frac{\partial L}{\partial \hat{x}_i^{(b)}} - \frac{1}{m} \sum_{b'=1}^{m} \frac{\partial L}{\partial \hat{x}_i^{(b')}} - \frac{\hat{x}_i^{(b)}}{m} \sum_{b'=1}^{m} \frac{\partial L}{\partial \hat{x}_i^{(b')}} \hat{x}_i^{(b')} \right).$$

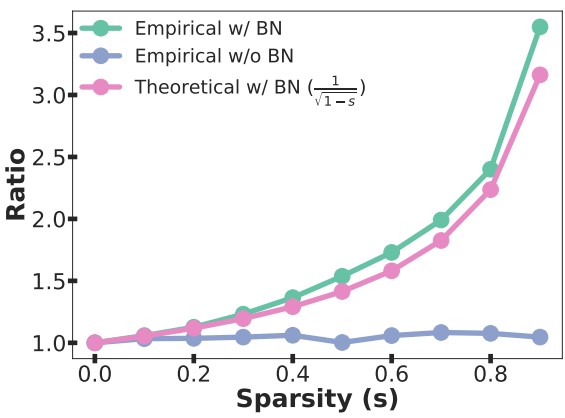

*Figure 2.* **Theoretical vs. empirical effect of Batch Normalization on gradients**. As observed theoretically in Section 3, the gradients of a sparse layer with BN can increase with sparsity, leading to instability during training. Here we show how our analytical scaling matches that of real gradients in Equation (11).

This can then be written as:

$$\frac{\partial L}{\partial x_i^{(b)}} = \frac{1}{\sigma_i} C_i^{(b)}, \tag{9}$$

where $C_i^{(b)}$ is defined as:

$$C_i^{(b)} := \left( \frac{\partial L}{\partial \hat{x}_i^{(b)}} - \frac{1}{m} \sum_{b'} \frac{\partial L}{\partial \hat{x}_i^{(b')}} - \frac{\hat{x}_i^{(b)}}{m} \sum_{b'} \frac{\partial L}{\partial \hat{x}_i^{(b')}} \hat{x}_i^{(b')} \right).$$

Note that $C_i^{(b)}$ depends only on the normalized activations $\hat{x}_i^{(b)}$ and their gradients. Since $\hat{x}_i^{(b)}$ is normalized to unit variance, $C_i^{(b)}$ is independent of the scale of the original pre-activations and therefore independent of $\sigma_i$ and the sparsity level.

For a sparse neuron with $\sigma_i' = \sqrt{1-s_i}\,\sigma_i$, substitution into (9) gives

$$\frac{\partial L}{\partial x_i'^{(b)}} = \frac{1}{\sigma_i'} C_i^{(b)} = \frac{1}{\sqrt{1-s_i}} \frac{\partial L}{\partial x_i^{(b)}}. \tag{10}$$

Thus, sparsity amplifies gradients by a factor $(1-s_i)^{-1/2}$, while the dense case ($s_i = 0$) yields unit scaling.

**Implication.** Equation (10) shows that sparsity amplifies gradients by a factor $(1-s_i)^{-1/2}$. In particular, the dense case ($s_i = 0$) recovers unit scaling, whereas increasing sparsity induces progressively larger updates. We empirically validate this for a two layer feed-forward network in Figure 2 (see Appendix E.3 for details).

**Gradient scaling under sparsity.** For a neuron $i$ with sparsity $s_i$,

$$\frac{\partial L}{\partial x_i'} = \frac{1}{\sqrt{1-s_i}} \frac{\partial L}{\partial x_i}. \tag{11}$$

BN amplifies gradients of sparse neurons by a factor $(1-s_i)^{-1/2}$, while dense ($s_i = 0$) recovers unit scaling.

### 3.3. How does Batch Normalization affect DST?

DST methods periodically update/change the network topology by updating the binary mask during training. At every $\Delta T$ steps, a subset of active connections is pruned — typically according to weight magnitude. To keep the sparsity constant, equivalent number of connections are regrown; SET (Mocanu et al., 2018) regrows connections randomly while RigL regrows connections using the (dense) gradient magnitude (Evci et al., 2020). Consequently, the sparsity pattern of the network changes abruptly at each mask update, and the neuron-wise sparsity levels ($s_i$) may change at each mask update step.

As shown in Section 3, the BN backward pass rescales the pre-activation gradient of neuron $i$ by the factor $(1-s_i)^{-1/2}$, as shown in Figure 1. Therefore, the effective gradient magnitude of each neuron depends explicitly on its specific sparsity ($s_i$). These insights suggest that our observations on how BN affects sparse layers may be exacerbated by mask updates in DST — whenever the mask is updated, the sparsity $\{s_i\}$ changes abruptly, which in turn induces an abrupt and neuron-dependent rescaling of some of the components of the gradient. Furthermore, this rescaling can be non-uniform across neurons as different neurons generally have different sparsities.

Such non-uniform rescaling *alters the gradient direction*, not only the gradient magnitude. Specifically, multiplying each neuron's gradient component by a different factor changes their relative contributions, thereby rotating the update direction in parameter space. Consequently, the optimizer observes a discontinuous change in the descent direction immediately after each DST mask update.

This effect introduces additional optimization noise to the training trajectory, and can lead to training instability and slower convergence in DST. In contrast, the dense case ($s_i = 0\ \forall i$) exhibits uniform scaling and does not suffer from this issue. To the best of our knowledge, this work is the first to identify BN as a source of such instability in DST and sparse training in general.

**BN–DST interaction.** DST mask updates change neuron-wise sparsity $s_i$, and BN rescales gradients by $(1-s_i)^{-1/2}$. The resulting non-uniform scaling alters gradient direction and can destabilize training.

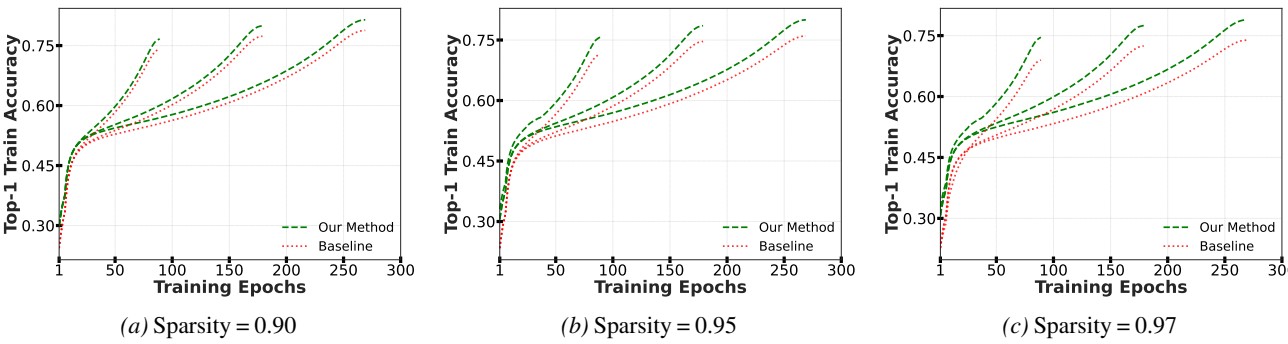

*(a)* Sparsity = 0.90          *(b)* Sparsity = 0.95          *(c)* Sparsity = 0.97

*Figure 3.* **Train accuracy (top-1) vs. epochs on ImageNet with RigL**. As observed, our method significantly improves the training dynamics and convergence of RigL, especially for higher sparsities.

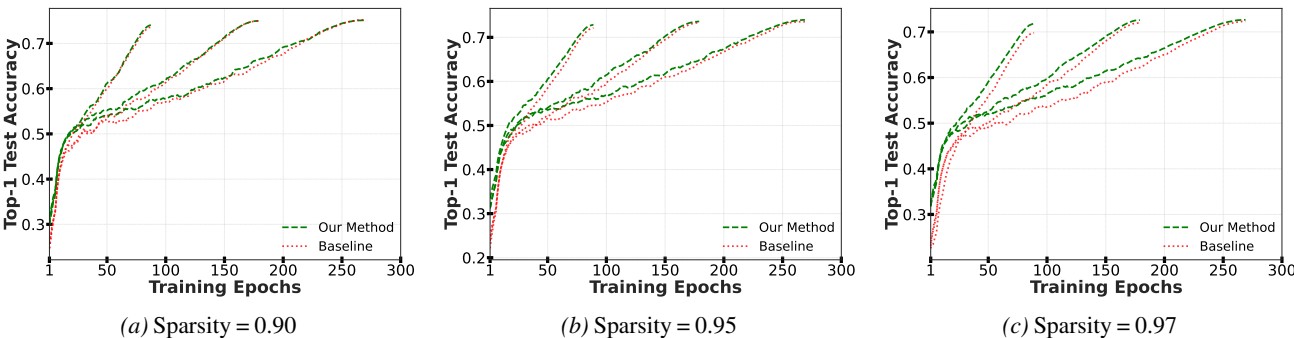

*(a)* Sparsity = 0.90          *(b)* Sparsity = 0.95          *(c)* Sparsity = 0.97

*Figure 4.* **Test accuracy (top-1) vs. epochs on ImageNet with RigL**. Our method converges faster and achieves higher accuracy with fewer training epochs, particularly at higher sparsity levels. With much longer training schedules, both methods converge to similar final accuracy. Detailed results can be found in Table 1.

## 4. Methodology

As discussed in Section 3.3, due to normalization layers, the direction of the gradients are altered for unstructured sparse layers. In DST methods specifically, the gradient direction can be changed abruptly during the mask update step; likely leading to unstable training dynamics. Using this insight, we propose a sparsity-aware preconditioned gradient descent optimizer, **SparseOpt**, that takes into account the sparsity distribution of the layer/neurons. The preconditioned matrix transforms the gradients to maintain the direction during training, which improves the sparse training dynamics and convergence.

### 4.1. Sparsity-Aware Preconditioned Gradient Descent

From Equation (10), the BN backward pass scales the pre-activation gradient of neuron $i$ by the factor $(1-s_i)^{-1/2}$, where $s_i \in [0, 1)$ denotes the neuron-wise sparsity level. Consequently, neurons with higher sparsity receive systematically larger gradients, introducing a neuron-dependent distortion in gradient magnitudes. This distortion is purely multiplicative and depends only on $s_i$. Therefore, it can be corrected by rescaling each neuron's gradient ($\nabla_i$) by the inverse factor $\sqrt{1-s_i}$, which restores uniform scaling across

neurons. In other words, if BN induces:

$$\nabla_i \propto (1-s_i)^{-1/2},$$

we apply the correction:

$$\nabla_i \leftarrow \sqrt{1-s_i}\nabla_i,$$

so that the effective gradient scale becomes independent of sparsity. We implement/formalize this correction using a sparsity-aware diagonal preconditioner $D$. For a weight matrix $\mathbf{W}$, the column $\mathbf{W}[:,i]$ corresponding to all incoming weights of neuron $i$ is scaled by $\sqrt{1-s_i}$. Equivalently, $\mathbf{D}$ is a diagonal matrix whose entries encode the neuron-wise factors $\sqrt{1-s_i}$. To preserve the overall gradient magnitude, we additionally normalize by $\sqrt{1-s_{\mathrm{avg}}}$, where $s_{\mathrm{avg}}$ denotes the average sparsity across neurons.

---

**SparseOpt update rule.** The resulting sparsity-aware preconditioned gradient descent update is:

$$w^{t+1} = w^t - \frac{\eta}{\sqrt{1-s_{\mathrm{avg}}}}D\nabla\mathcal{L}(w^t), \qquad (12)$$

---

where $\eta$ is the learning rate. This update can be interpreted as a diagonal preconditioning step that corrects the neuron-wise scaling induced by BN while approximately preserving the

global gradient norm. In Section 5, we show empirically that this correction removes the BN-induced gradient imbalance and improves optimization stability in DST.

**Layer with homogeneous connectivity (e.g., dense).** Consider a layer in which all neurons have identical (homogeneous) connectivity, so that $s_i = 0$ for all $i$. The correction factors therefore satisfy $\sqrt{1-s_i} = 1$, implying that the preconditioner reduces to the identity, $\mathbf{D} = \mathbf{I}$. Consequently, equation (12) becomes exactly the standard SGD update. Thus, dense training with SGD is a special case of our general preconditioned formulation, which is compatible with heterogeneous and sparse layers.

## 5. Results

We validate our hypothesis on the CIFAR-100 (Krizhevsky et al., 2009) and ImageNet/ILSVRC2012 (Deng et al., 2009) datasets using ResNet architectures (He et al., 2016) across multiple sparsity levels ($s \in \{0.90, 0.95, 0.97\}$) in Section 5.1. We use RigL (Evci et al., 2020) and SET (Mocanu et al., 2018), as our DST methods. For each method, we compare standard SGD w/ momentum with our proposed sparsity-aware preconditioned optimizer (a sparsity-aware optimizer (SparseOpt)). This comparison isolates the effect of correcting the BN-induced gradient skew caused by heterogeneous sparsity. Since RigL regrows connections based on the gradient magnitude, we only use the corrected gradients, as in Equation (12) for updating weights; we observe that using original gradients for regrowing achieves better generalization. We provide more analysis in Section 5.2. We empirically show that SparseOpt, by accounting for heterogeneous connectivity and sparsity of layers through the proposed preconditioning, consistently improves convergence and generalization — enabling models to reach higher accuracy with shorter training schedules (fewer total epochs). To further understand the effect of BN on DST, we analyze the mask exploration rate of RigL with our proposed method against the baseline to quantitatively measures how much sparse topologies are explored during training in Section 5.2.

### 5.1. Experimental Details

**ResNet50/ImageNet.** We train RigL and SET across multiple training schedules (totalling 90, 180, and 270 epochs) to evaluate whether SparseOpt improves the convergence and generalization of DST methods under limited training budgets, compared to standard SGD. For a fair comparison, we perform a hyperparameter sweep over the learning rate, drop fraction, and drop frequency for both the baselines and the proposed method (Appendix E). Figure 3 shows the training curves for RigL across different schedules. Our method consistently accelerates convergence, particularly at higher sparsities where connectivity is more heterogeneous.

*Table 1.* **Test Acc. (Top-1) on ImageNet/ResNet50**. SparseOpt consistently achieves better generalization than baseline BN across both SET (Mocanu et al., 2018) and RigL (Evci et al., 2020) esp. with smaller training budgets.

| Sparsity | DST | Method | Training Epochs | | |
|---|---|---|---|---|---|
| | | | 90 | 180 | 270 |
| 90% | RigL | Ours | **74.41** | **75.06** | **75.26** |
| | | Baseline | 73.88 | 74.99 | **75.26** |
| | SET | Ours | **74.15** | 74.81 | 74.85 |
| | | Baseline | 73.69 | **74.83** | **75.17** |
| 95% | RigL | Ours | **72.93** | **73.62** | **73.95** |
| | | Baseline | 72.33 | 73.36 | 73.62 |
| | SET | Ours | **72.59** | **73.75** | 73.79 |
| | | Baseline | 71.84 | 73.51 | **73.82** |
| 97% | RigL | Ours | **71.15** | **72.65** | **72.66** |
| | | Baseline | 69.94 | 72.00 | 72.35 |
| | SET | Ours | **71.43** | **72.40** | **72.73** |
| | | Baseline | 70.12 | 72.17 | 72.65 |

Figure 4 shows the corresponding test accuracy, where SparseOpt achieves higher generalization using fewer training epochs, while it requires extended training for baseline to converge to similar accuracy. We show detailed results for RigL in Table 1, observing significant improvement in top-1 accuracy over the baseline. We also validated our method with SET, leading as well to consistently improved model generalization with fewer training epochs as shown in Table 1. We observed that RigL performs better than SET, showing that RigL explores sparse topologies better using gradient information, compared to random exploration in SET

**ResNet20/CIFAR-100.** We also validated our method on the CIFAR-100 dataset (Krizhevsky et al., 2009) using both RigL and SET as baselines across different high sparsity regimes ($s \in \{0.90, 0.95, 0.97\}$) for different training schedules ($100, 200, 300, 500$ epochs). We perform a grid search to find the best hyperparameters for both the baselines and our method for a fair comparison (Appendix E). We show the results in Table 2 for different training schedules and sparsities trained with RigL; our proposed method consistently outperforms the baseline, especially when trained with a shorter schedule. This empirically shows that our method improves the convergence for DST. We observe similar improvement in convergence when trained with SET, as shown in Table 3. In Table 2 and Table 3, we also compare our method with the current state-of-the-art sparse optimizer, HAM (Jacobs et al., 2026); we present more details in Section 5.3. Our results empirically demonstrate that our proposed method improves DST convergence across different datasets, enabling them to achieve higher accuracy with shorter training schedules. We show training and test accuracy evolution for CIFAR-100 in Appendix C.3.

*Table 2.* **Test accuracy on CIFAR-100 dataset with RigL**. Our proposed method improves the convergence across different sparsity levels and achieves higher generalization; the baseline takes many more training epochs to converge to similar accuracy.

| Sparsity | Method | Training Epochs | | | |
|---|---|---|---|---|---|
| | | 100 | 200 | 300 | 500 |
| 90% | Ours (w/ HAM) | **61.64±0.24** | **63.37±0.50** | **64.44±0.55** | 64.94±0.25 |
| | Ours (w/o HAM) | 61.34±0.35 | 63.12±0.39 | 64.30±0.08 | **65.22±0.40** |
| | Baseline (w/ HAM) | 60.80±0.51 | 62.89±0.13 | 64.21±0.07 | 64.89±0.27 |
| | Baseline | 60.66±0.43 | 62.93±0.17 | 63.77±0.35 | 64.66±0.17 |
| 95% | Ours (w/ HAM) | **57.31±0.54** | **60.41±0.28** | **60.91±0.45** | **62.20±0.38** |
| | Ours (w/o HAM) | 56.80±0.49 | 59.91±0.45 | 60.67±0.63 | 61.74±0.35 |
| | Baseline (w/ HAM) | 56.72±0.40 | 58.76±0.60 | 60.38±0.23 | 61.66±0.26 |
| | Baseline | 55.80±0.73 | 59.22±0.55 | 60.00±0.54 | 61.45±0.65 |
| 97% | Ours (w/ HAM) | **54.06±0.10** | 56.30±0.28 | 57.48±0.82 | **59.13±0.92** |
| | Ours (w/o HAM) | 53.95±0.40 | **57.01±0.19** | **57.62±0.59** | 58.35±0.34 |
| | Baseline (w/ HAM) | 52.74±0.69 | 54.58±0.67 | 56.45±0.13 | 57.82±0.76 |
| | Baseline | 52.53±0.35 | 54.87±0.78 | 56.50±0.33 | 57.32±0.16 |

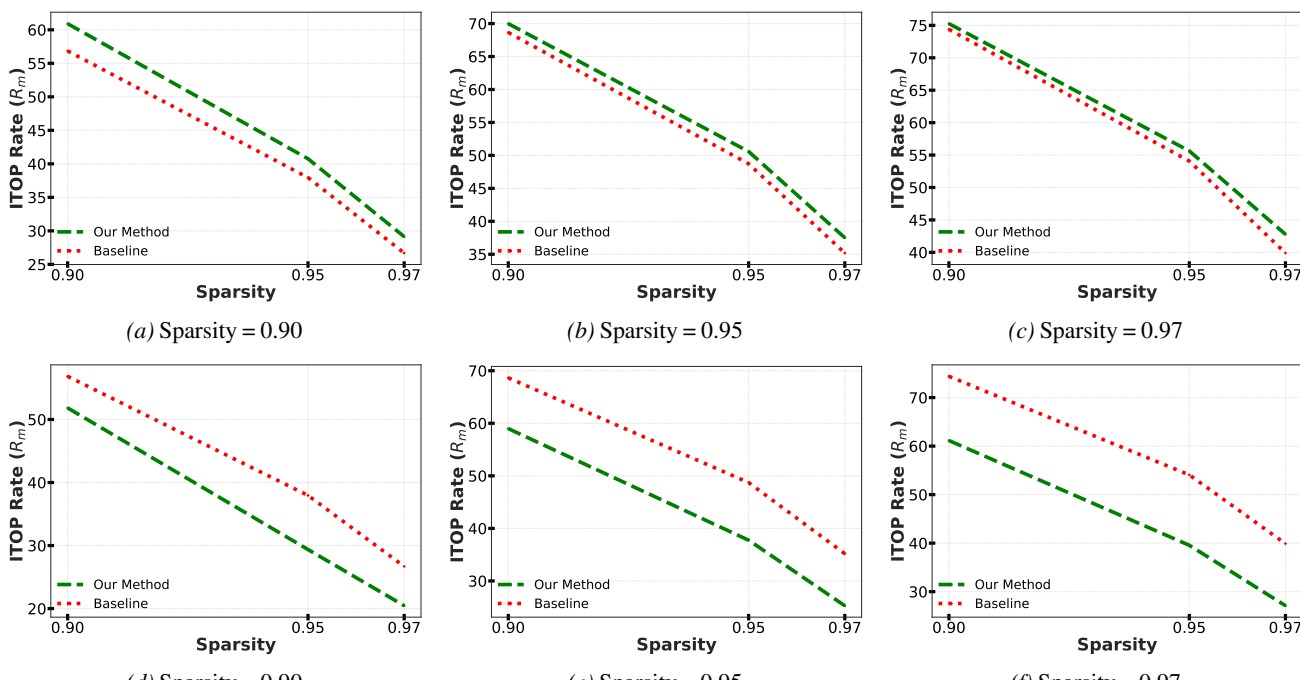

*(a)* Sparsity = 0.90      *(b)* Sparsity = 0.95      *(c)* Sparsity = 0.97

*(d)* Sparsity = 0.90      *(e)* Sparsity = 0.95      *(f)* Sparsity = 0.97

*Figure 5.* **RigL ITOP rate ($R_m$) vs. sparsity for ResNet50/ImageNet**. The top row shows results when RigL uses the original gradients for mask exploration, while the bottom row uses the corrected gradients. Differences in ITOP indicate that BN through scaling of gradients influences which connections are regrown and thus affects mask exploration.

### 5.2. How does BN affect mask exploration?

One key difference between DST and static sparse training, which uses a constant mask throughout training, is that DST updates the sparse mask at regular intervals. This allows DST to explore different sparse topologies, helping it find a good subspace. As a result, DST can match the performance of dense models, unlike static sparse training. Liu et al. (2021) proposed that sufficient mask exploration, i.e. exploring a diverse set of topologies, is required to

match dense performance. To quantitatively measure mask exploration, Liu et al. (2021) introduced the ITOP rate ($R_m$), defined as the union of all masks encountered during training, normalized by the total number of parameters.

DST mask exploration depends on both the pruning criterion, which is typically based on the lowest-magnitude parameters, and the regrowth criterion, which is random for SET and gradient-magnitude–based for RigL. Since regrowth in RigL relies explicitly on gradient magnitude, any modification

to the gradients can directly influence mask exploration. To understand the effect of BN on mask exploration, we analyze the ITOP rate of the RigL baseline together with our proposed method. Because our method alters the gradients used for weight updates, it may also affect which connections are selected during regrowth. Therefore, we compare two settings: (i) using the corrected gradients, as in Equation (12), for mask updates and (ii) using the original gradients for mask exploration.

As observed in Figure 5, using the corrected gradients only for the weight updates improves the ITOP rate compared to the baselines. This highlights the interplay between BN and mask exploration. While the corrected gradients help improve training dynamics, using the original gradients is more effective for mask exploration. This could be explained by the fact that the gradient magnitude in RigL directly determines which connections are regrown; modifying the gradients alters this ranking and can bias the regrowth process, thereby reducing the diversity of explored masks. Thus, mask exploration in DST is not independent of optimization — through gradient rescaling, BN implicitly steers which connections are regrown and which sparse topologies are discovered.

### 5.3. Compatibility with other optimizers

Recent works have proposed optimizers specifically for sparse training to improve the convergence rate (Liu et al., 2021; Lei et al., 2024; Ji et al., 2024). However, none of the previous work considers the effect of normalization layers on the sparse training dynamics. We show that SparseOpt is complementary with one such optimizer: Hyperbolic Aware Minimization (HAM) (Jacobs et al., 2026) — currently the state-of-the-art — in Appendix A.1. We prove in a one-neuron example that the mechanisms are orthogonal to each other, captured by Lemma A.3. Therefore, our method can be used along with the other optimization methods to further improve the rate of convergence. We empirically validate this on the CIFAR-100 dataset in Figures 17 and 18 in Appendix C.4.

## 6. Conclusion

Normalization layers — now ubiquitous in all modern DNNs — have significantly improved training dynamics and enabled the optimization of deeper and wider architectures. While their role in dense training has been extensively studied, their effect and efficacy in sparse training regimes remain largely unexplored. In this work, we showed that BN adversely affects sparse training, and proposed SparseOpt, a new sparsity-aware optimization method to improve the convergence of DST methods. We show empirically that SparseOpt consistently improves convergence and generalization across sparsity levels with minimal computational overhead and straightforward implementation. To the best of our knowledge, this is the first work to highlight the need for sparsity-aware nor-

malization mechanisms to enable practical sparse training. Our work highlights the limitations of normalization layers for sparse training, and to the best of our knowledge, this is the first work to highlight the need for sparsity-aware normalization mechanisms to enable practical sparse training.

Our findings also suggest that many existing training methodologies implicitly assume uniform or homogeneous connectivity across neurons — an assumption that holds in dense layers but is violated in sparse or otherwise heterogeneously-connected layers. When this uniform connectivity assumption is violated, even commonly used training methodologies such as normalization can behave in unexpected ways, highlighting the need for sparsity-aware design rather than directly transferring techniques developed for dense models.

**Future Work.** A natural next step is to extend our analysis to other normalization layers, which may enable more stable and improve the state of sparse training for transformer-based models. Our preliminary analysis shows that Layer Normalization also adversely affects sparse training, as shown in Appendix D. Future work will focus on correcting the gradient skew induced by Layer Normalization (LN) and LLM sparse training. However, this will be a focus of future study and out of scope for this work.

**Limitations.** While our work significantly improves DST convergence rates, the effectiveness of DST methods depends on the interplay between training dynamics and mask-exploration strategies. However, their coupled effects make it difficult to disentangle their respective contributions to convergence. In this work, we focus exclusively on the role of normalization layers, providing novel insights into their impact on DST. Developing a deeper understanding of mask exploration and designing improved strategies for it remains an important direction for future research. Although our work does not fully resolve the challenges of DST, it offers new insights that we believe will be valuable to the sparse-training community and will help motivate future research on sparse training methods.

## Impact Statement

This work studies the role of normalization layers in sparse training and proposed a method to improve the state of sparse training. By enabling sparse models to better match dense performance, our approach can reduce the computational and memory requirements of training and deployment. Such improvements may lower energy consumption, decrease hardware costs, and make large-scale deep learning more accessible to researchers and practitioners with limited resources, thereby contributing to more sustainable and democratized machine learning. Our work is algorithmic in nature and does not target any specific application domain; therefore, these risks stem from downstream uses rather than the method itself. We encourage future work to combine advances in efficient training with responsible deployment practices, including careful evaluation of fairness, robustness, and environmental costs.

## Acknowledgement

MA is supported by the NSERC Postgraduate Scholarship, RBC Borealis through the Borealis AI Global Fellowship Award, Killam Doctoral Fellowship, and the Digital Research Alliance of Canada EDIA Champions program. RGK is supported by a Canada CIFAR AI Chair and a Canada Research Chair Tier II in Computational Medicine (CRC-2022-00049). RB gratefully acknowledges the Gauss Centre for Supercomputing e.V. (www.gauss-centre.eu) for funding this project by providing computing time on the GCS Supercomputer JUWELS at Jülich Supercomputing Centre (JSC). RB is also grateful for funding from the European Research Council (ERC) under the Horizon Europe Framework Programme (HORIZON) for proposal number 101116395 SPARSE-ML.

We gratefully acknowledge the support of Alberta Innovates (ALLRP 577350-22, ALLRP 600038-24), the Natural Sciences and Engineering Research Council of Canada (NSERC) (RGPIN-2022-03120, DGECR-2022-00358), Defence Research and Development Canada (DGDND-2022-03120), and NSERC/Agence Nationale de la Recherche (ANR) (ALLRP 602719-24). This project was undertaken thanks to funding from IVADO and the Canada First Research Excellence Fund. This research was enabled in part by support provided by the Digital Research Alliance of Canada (alliancecan.ca). Resources used in preparing this research were provided, in part, by the Province of Ontario, the Government of Canada through CIFAR, and companies sponsoring the Vector Institute.

## Contribution Statement

All the authors contributed to the writing of the paper. YI and MA conceived initial research direction. MA derived the methodology and effect of normalization layers on sparse training dynamics and implemented the method. RJ contributed in running experiments and preparing the manuscript. TJ contributed in the derivation of HAM optimizer (Section A.1). ES contributed to training stability analysis for the earlier version of the manuscript along with feedback on the methodology. RGK, RB, and YI, as senior authors, provided feedback on the writing and methodology. YI designed Figure 1, and also contributed feedback on experiments throughout the project.

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

# A. Appendix

## A.1. Scaling is compatible with the HAM optimizer

In this section we show how the optimizer HAM (Jacobs et al., 2026) is compatible with our BN rescaling. We adopt the two layer neural network setting as in (Gadhikar et al., 2025) to prove that HAM allows for the correct sign flip, with the main difference being now that we include BN in the flow and scaling. This is achieved by deriving invariants of the flow of both gradient flow and the induced HAM flow. The crucial observation is that the scaling and masking affects only the flow of the inner group of parameters thus respecting the invariant between the outer layer parameter and the BN parameters. Note to keep the analysis concise we omit the effect of the weight decay or HAM regularization as used in (Jacobs & Burkholz, 2025; Jacobs et al., 2025).

**Setup:** Consider a two-layer neural network $f : \mathbb{R}^{n+1} \times \mathbb{R}^n \to \mathbb{R}$ with full BN under gradient flow and balanced initialization. Moreover, denote the data set by $(z_j, \hat{y}_j) \in \mathbb{R}^{n+1}$ for $j \in [m]$ with $[m] := \{1, ..., m\}$. Note that now the balance applies to the BN and outer layer parameter not the weights of the inner layer. Concretely the network is represented as:

$$f(w, a | z) := a\sigma(w^T z) \tag{13}$$

where $\sigma(\cdot) := \max\{0, \cdot\}$ is the rectified linear unit (Positive homogeneous activation). The BN operation acting on the pre-activations $X := \{x_j\}_{j \in [m]} \in \mathbb{R}$ is defined as:

$$BN(x_j, \beta, \gamma) := \gamma \frac{x_j - \mu_X}{\sqrt{\sigma_X^2 + \epsilon}} + \beta$$

where $\mu_X := \frac{1}{m} \sum_{j \in [m]} x_j$ and $\sigma_X^2 := \frac{1}{m} \sum_{j \in [m]} (x_j - \mu_X)^2$. Note we now use the subscript $X$ instead of $B$ to indicate that it is the full data set. We train with the mean squared error loss function, the objective becomes:

$$L(w, a | z) := \frac{1}{2m} \sum_{j \in [m]} (\hat{y}_j - a\sigma(wz_i))^2$$

First we recall the additional variables $\hat{x}_j := \frac{x_j - \mu_X}{\sqrt{\sigma_X^2 + \epsilon}}$ and $y_j := \gamma \hat{x}_j + \beta$. This allows use to write the gradient flow equation more clear for the invariant equation proof.

**Gradient flow:** In this setting, we can consider a gradient flow of the parameters described by a dynamical system of the form:

$$\begin{cases} da_t = \frac{1}{m} \sum_{j \in [m]} (\hat{y}_j - a_t \sigma(y_{j,t})) \, \sigma(y_{j,t}) dt, & a_0 = a_{\text{init}} \\ dy_{j,t} = \hat{x}_{j,t} d\gamma_t + \gamma_t d\hat{x}_{j,t} + d\beta_t \\ d\hat{x}_{j,t} = \frac{dx_{j,t} - d\mu_X}{\sqrt{\sigma_X^2 + \epsilon}} - \frac{x_{j,t} - \mu_X}{2(\sigma_X^2 + \epsilon)^{3/2}} d\sigma_X^2, \\ dw_t = \frac{1}{m} \sum_{j=1}^m (\hat{y}_j - a_t \sigma(y_{j,t})) a_t \sigma'(y_{j,t}) \gamma_t \cdot \\ \left( \frac{x_j - \frac{1}{m} \sum_{i=1}^m x_i}{\sqrt{\sigma_X^2 + \varepsilon}} - \frac{x_{j,t} - \mu_X}{m(\sigma_X^2 + \epsilon)^{3/2}} \sum_{i=1}^m (x_{i,t} - \mu_X)(z_i - \frac{1}{m} \sum_{\ell=1}^m z_\ell) \right) dt, & w_0 = w_{\text{init}}, \\ d\gamma_t = \frac{1}{m} \sum_{j \in [m]} (\hat{y}_j - a_t \sigma(y_{j,t})) a_t \sigma'(y_{j,t}) \hat{x}_{j,t} dt, & \gamma_0 = \gamma_{\text{init}} \\ d\beta_t = \frac{1}{m} \sum_{j \in [m]} (\hat{y}_j - a_t \sigma(y_{j,t})) a_t \sigma'(y_{j,t}) dt, & \beta_0 = \beta_{\text{init}} \end{cases}$$

**Lemma A.1.** *The following equation is invariant under the gradient flow:*

$$a_t^2 - \gamma_t^2 - \beta_t^2 = a_0^2 - \gamma_0^2 - \beta_0^2$$

*for $t \geq 0$.*

Proof. It directly follows from writing out the evolution. Notice that $y_j = \gamma \hat{x}_j + \beta$ and using gradient flow:

$$
\begin{aligned}
d(a_t^2 - \gamma_t^2 - \beta_t^2) &= 2a_t da_t - 2\gamma_t d\gamma_t - 2\beta_t d\beta_t \\
&= 2a_t \frac{1}{m} \sum_{j \in [m]} (\hat{y}_j - a_t \sigma(y_{j,t})) \, \sigma(y_{j,t}) dt - 2\gamma_t \frac{1}{m} \sum_{j \in [m]} (\hat{y}_j - a_t \sigma(y_{j,t})) a_t \sigma'(y_{j,t}) \hat{x}_{j,t} dt \\
&\quad - 2\beta_t \frac{1}{m} \sum_{j \in [m]} (\hat{y}_j - a_t \sigma(y_{j,t})) a_t \sigma'(y_{j,t}) dt \\
&= \frac{1}{m} \sum_{j \in [m]} (\hat{y}_j - a_t \sigma(y_{j,t})) \, (2a_t \sigma(y_{j,t}) - 2a_t y_{j,t} \sigma'(y_{j,t})) dt = 0.
\end{aligned}
$$

where we used that the ReLu is positive homogeneous i.e. $\sigma(x) = x\sigma'(x)$ for all $x \in \mathbb{R}$. This concludes the proof. $\square$

Lemma A.1 implies that if we initialize with $a^2 \leq \gamma^2$ and $\beta = 0$, $a$ can not sign flip. This includes the balanced initialization $a_0^2 = \gamma_0^2 + \beta_0^2$. Clearly, if we now apply a mask to $w$ and rescale this does not affect the invariant.

This is summarized by the following observation.

*Observation* A.2. The scaling and mask only change the gradient flow of $w_t$. Therefore, it does not affect the invariance in Lemma A.1.

**HAM gradient flow**  Consider now the HAM gradient flow which is a Riemannian gradient flow with inverse metric $g^{-1}(w) = 1 + \alpha|w|$, where $\alpha > 0$ is a hyperparameter. As in (Jacobs et al., 2026), the metric for BatchNorm is not changed. This changes the gradient flow system to:

$$
\begin{cases}
da_t = (1 + \alpha|a_t|) \frac{1}{m} \sum_{j \in [m]} (\hat{y}_j - a_t \sigma(y_{j,t})) \, \sigma(y_{j,t}) dt, & a_0 = a_{\text{init}} \\
dy_{j,t} = \hat{x}_{j,t} d\gamma_t + \gamma_t d\hat{x}_{j,t} + d\beta_t \\
d\hat{x}_{j,t} = \dfrac{dx_{j,t} - d\mu_X}{\sqrt{\sigma_X^2 + \epsilon}} - \dfrac{x_{j,t} - \mu_X}{2(\sigma_X^2 + \epsilon)^{3/2}} d\sigma_X^2, \\
dw_t = (1 + \alpha|w_t|) \odot \frac{1}{m} \sum_{j=1}^m (\hat{y}_j - a_t \sigma(y_{j,t})) a_t \sigma'(y_{j,t}) \gamma_t \cdot \\
\quad \left( \dfrac{z_j - \frac{1}{m}\sum_{i=1}^m z_i}{\sqrt{\sigma_X^2 + \epsilon}} - \dfrac{x_{j,t} - \mu_X}{m(\sigma_X^2 + \epsilon)^{3/2}} \sum_{i=1}^m (x_{i,t} - \mu_X)(z_i - \frac{1}{m}\sum_{\ell=1}^m z_\ell) \right) dt, & w_0 = w_{\text{init}}, \\
d\gamma_t = \frac{1}{m} \sum_{j \in [m]} (\hat{y}_j - a_t \sigma(y_{j,t})) a_t \sigma'(y_{j,t}) \hat{x}_{j,t} dt, & \gamma_0 = \gamma_{\text{init}} \\
d\beta_t = \frac{1}{m} \sum_{j \in [m]} (\hat{y}_j - a_t \sigma(y_{j,t})) a_t \sigma'(y_{j,t}) dt, & \beta_0 = \beta_{\text{init}}
\end{cases}
$$

We can again proof an invariant of the flow.

**Lemma A.3.** *The following equation holds for HAM gradient flow:*

$$
\int^{a_t} \frac{p}{1 + \alpha|p|} dp - \frac{1}{2}(\gamma_t^2 + \beta_t^2) = \int^{a_0} \frac{p}{1 + \alpha|p|} dp - \frac{1}{2}(\gamma_0^2 + \beta_0^2)
$$

*for $t \geq 0$. The integral is equal to*

$$
\int^x \frac{p}{1 + \alpha|p|} dp = \frac{\alpha|x| - \ln(|\alpha|x| + 1|)}{\alpha^2}
$$

Proof. The proof is similar to Lemma A.1. We differentiate the left hand side with respect to the flow.

$$d\left(\int^{a_t} \frac{p}{1+\alpha|p|}dp - \frac{1}{2}\left(\gamma_t^2 + \beta_t^2\right)\right) = \frac{a_t}{1+\alpha|a_t|}da_t - \gamma_t d\gamma_t - \beta_t d\beta_t$$

$$= a_t\frac{1}{m}\sum_{j\in[m]}\left(\hat{y}_j - a_t\sigma(y_{j,t})\right)\sigma(y_{j,t})dt - \gamma_t\frac{1}{m}\sum_{j\in[m]}(\hat{y}_j - a_t\sigma(y_{j,t}))a_t\sigma'(y_{j,t})\hat{x}_{j,t}dt$$

$$- \beta_t\frac{1}{m}\sum_{j\in[m]}(\hat{y}_j - a_t\sigma(y_{j,t}))a_t\sigma'(y_{j,t})dt$$

$$= \frac{1}{m}\sum_{j\in[m]}(\hat{y}_j - a_t\sigma(y_{j,t}))\left(a_t\sigma(y_{j,t}) - a_t y_{j,t}\sigma'(y_{j,t})\right)dt = 0.$$

Note that the main difference with Lemma A.1 is the cancellation of $1+\alpha|a_t|$. This concludes the result. □

**Corollary A.4.** *If $\gamma_0^2 + \beta_0^2 > 2\int^{a_0}\frac{p}{1+\alpha|p|}dp$, then $a$ can flip its sign*

Proof. For $a_t$ to sign flip its flow has to be able to move trough zero. Therefore, the balance equation needs to have a non zero solution for $a_t = 0$ i.e. not $\gamma_t = \beta_t = 0$. Plugging this in gives

$$-\frac{1}{2}\left(\gamma_t^2 + \beta_t^2\right) = \int^{a_0}\frac{p}{1+\alpha|p|}dp - \frac{1}{2}\left(\gamma_0^2 + \beta_0^2\right)$$

This has a non zero solution iff the right hand side is negative as the left hand side is non-positive. This concludes the proof. □

Note now that $a_t$ can sign flip now under the balanced initialization ($a_0^2 = \gamma_0^2 + \beta_0^2$) making it possible to recover the ground truth (Gadhikar et al., 2025). In Figures 6a and 6b we illustrate the invariance for a balanced initialization. Note that invariant for gradient flow becomes singular at $a = 0$, while HAM's invariant does not.

*Remark* A.5. These balance equations can be easily extended to the multi-neuron case. We can see this from the chain rule, as the neurons are summed up they do not interact with each others balance equation.

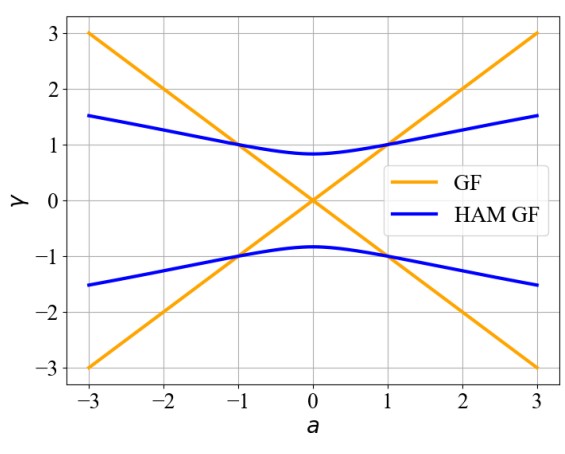

*(a)* A 2D display of the balance equation.

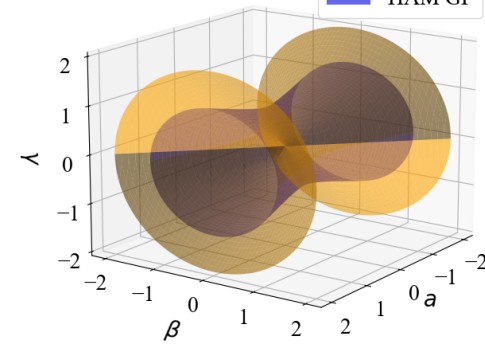

*(b)* A 3D display of the balance equation.

*Figure 6.* The 2D ($\beta = 0$) and 3D representations of the balance equation for $a$, $\gamma$ and $\beta$ initialized at balance for gradient flow. The $a$ parameter can flip its sign in case of HAM while this is not possible for balanced GF.

**Experimental simulation** We illustrate the consequences of the balance relationship in the presence of scaling. We consider a one neuron with multi-dimensional input and a mask.

**One neuron** We train a student neuron with input dimension 10 and output dimension 1 to learn a similar teacher neuron. To illustrate the sign flip we initialize them with opposite $a$ signs. The training data is generated by the teacher and inputs

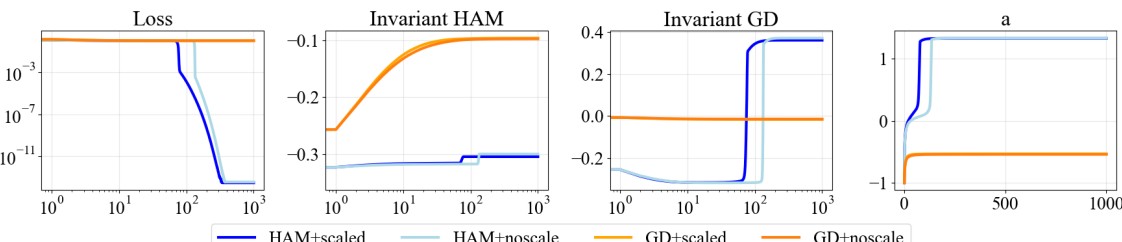

*Figure 7.* One neuron student teacher dynamics with $\eta = 0.01$, HAM is needed for a sign flip. Under small learning rate the convergence is similar independent of scaling.

*Figure 8.* One neuron student teacher dynamics with large learning rate $\eta = 0.1$, HAM is needed for a sign flip. The scaling allows for faster convergence.

are 200 i.i.d. samples from a Gaussian for each entry $N(0,1)$. We train with constant learning rate $\eta \in [0.01, 0.1]$ and train for 10000,1000 iterations, to ensure convergence. The BN parameters are initialized as $\gamma_0 = 1$ and $\beta_0 = 0$, this together with $a_0^2 = 1$ controls the balance equation. For HAM, we set $\alpha = 4$. The teacher has 8 redundant input features, for the student we mask these. This effectively mimics the late stage of dynamic sparse training where we have identified the correct mask.

In Figures 7 and 8, we observe that indeed the predicted invariances stay satisfied with scaling and masking. Furthermore, HAM converges with and without scaling to the ground truth while gradient descent fails. The scaling improves convergence especially for the larger learning rate. Highlighting the benefit of the scaling correction.

**Multi neuron** Here we conduct an experiment with two student neurons and the same teacher neuron with 8 redundant entries. Now we training with one dense neuron where all the parameters are active except the 2 parameters that are not redundant, and again a sparse neuron where the correct redundant entries are masked. We train with large learning rate $\eta = 0.1$ for 1000 iterations.

In Figure 9 we observe that the invariants are again preserved. Moreover, very different dynamics are now occurring for scaling or not scaling. If we rescale, the dense neuron is turned off i.e. $a_{\text{dense}}$ becomes small, while this is not the case in the other scenario. This indicates that scaling can help explore the parameter space better. In other words the scaling gives (correct) sparse neurons a chance to learn.

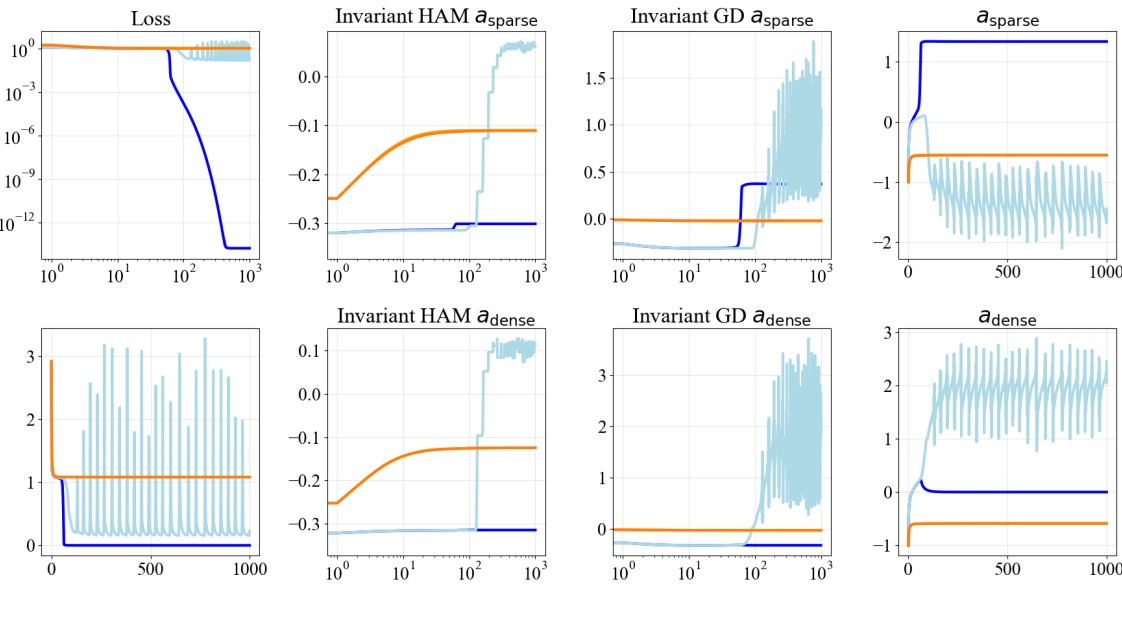

*Figure 9.* Multi neuron student with one dense and sparse neuron and learning rate $\eta = 0.1$. Again HAM is needed for the sign flip substantiating the balance equation. Rescaling can now lead to learning a different sparser representation with the dense (redundant) neuron turned off.

# B. Effect of BN on Mask Exploration

In this section, we provide additional experimental results on the RigL ITOP rate ($R_m$) vs. sparsity for CIFAR-100, as shown in Figure 10.

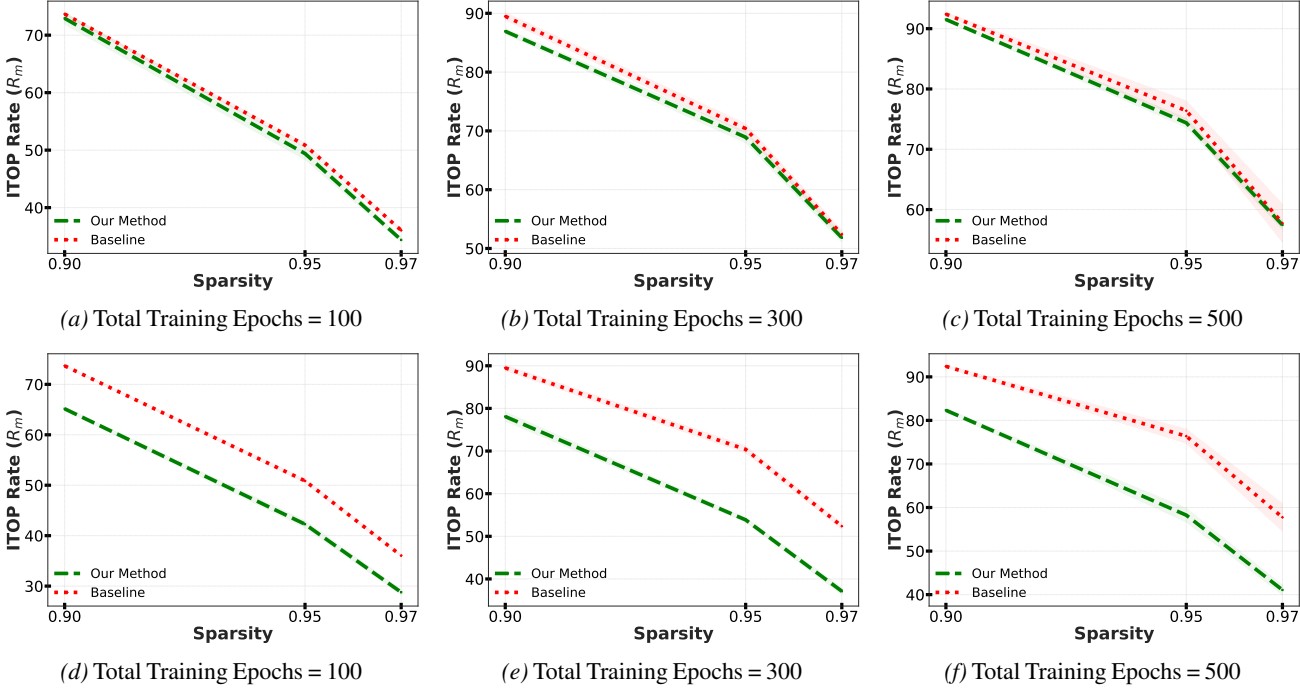

*(a)* Total Training Epochs = 100      *(b)* Total Training Epochs = 300      *(c)* Total Training Epochs = 500

*(d)* Total Training Epochs = 100      *(e)* Total Training Epochs = 300      *(f)* Total Training Epochs = 500

*Figure 10.* **RigL ITOP rate ($R_m$) vs. sparsity for ResNet20×{1}/CIFAR-100**. The top row shows results when RigL uses the original gradients for mask exploration, while the bottom row uses the corrected gradients. Differences in ITOP indicate that BN through scaling of gradients influences which connections are regrown and thus affects mask exploration.

# C. Additional Experiments

## C.1. Results

In this section, we present a detailed numerical comparison of test accuracy across various sparsity levels for CIFAR-100. These results demonstrate the consistent performance improvements of our proposed method over standard baselines, as shown in Table 3.

*Table 3.* **Test accuracy on CIFAR-100 dataset with SET**. Our proposed method improves the convergence across different sparsity levels and achieves higher generalization; the baseline takes many more training epochs to converge to similar accuracy.

| Sparsity | Method | Training Epochs | | | |
|---|---|---|---|---|---|
| | | 100 | 200 | 300 | 500 |
| 90% | Ours (w/ HAM) | **62.04±0.43** | **63.86±0.20** | **64.75±0.25** | 65.20±0.12 |
| | Ours (w/o HAM) | 61.97±0.20 | 63.55±0.17 | 64.09±0.25 | **65.21±0.31** |
| | Baseline (w/ HAM) | 60.99±0.11 | 63.00±0.44 | 64.19±0.21 | 65.07±0.30 |
| | Baseline | 60.84±0.50 | 63.11±0.25 | 63.99±0.41 | 65.01±0.37 |
| 95% | Ours (w/ HAM) | **58.33±0.40** | **60.84±0.25** | **61.28±0.20** | **62.69±0.29** |
| | Ours (w/o HAM) | 58.04±0.43 | 60.71±0.28 | 61.05±0.70 | 62.46±0.06 |
| | Baseline (w/ HAM) | 56.87±0.29 | 59.96±0.26 | 60.76±0.13 | 62.09±0.58 |
| | Baseline | 56.59±0.17 | 59.74±0.15 | 60.77±0.08 | 61.81±0.36 |
| 97% | Ours (w/ HAM) | 54.60±0.34 | **57.42±1.15** | **58.51±0.34** | 59.30±0.35 |
| | Ours (w/o HAM) | **54.76±0.39** | 56.50±0.06 | 58.08±0.57 | 59.43±0.53 |
| | Baseline (w/ HAM) | 53.70±0.46 | 56.04±0.30 | 56.80±0.63 | 59.16±0.71 |
| | Baseline | 53.10±0.48 | 56.23±0.61 | 57.09±0.54 | **59.58±0.46** |

## C.2. Comparison of Test Accuracy vs. Total Training Epochs Plots

In this section, we evaluate the generalization performance of our method against the baseline by training independent models across varying total training epochs. We demonstrate that our method consistently achieves better convergence efficiency and final accuracy across all evaluated sparsity levels, as shown in Figure 11, Figure 12, Figure 13, and Figure 14. **Note**: The x-axis represents different models trained independently for different number of total number of epochs.

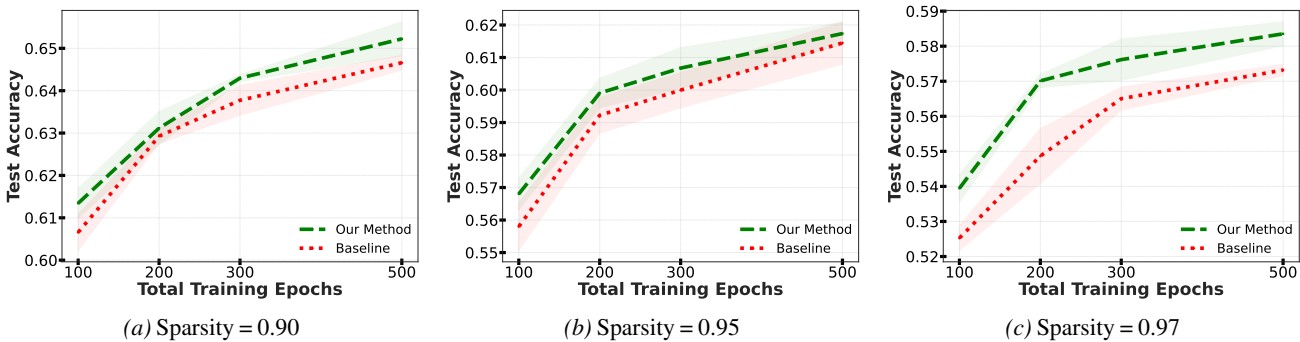

*(a)* Sparsity = 0.90          *(b)* Sparsity = 0.95          *(c)* Sparsity = 0.97

*Figure 11.* **Test accuracy vs. total training epochs for ResNet20×{1}/CIFAR-100 with RigL**. Our method consistently outperforms the baseline across all sparsity levels, achieving higher generalization accuracy and demonstrating improved rate of convergence.

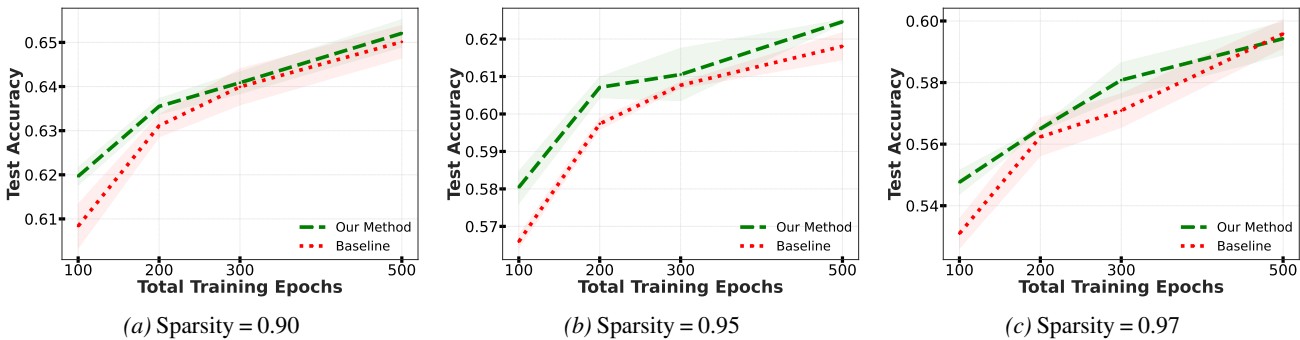

*Figure 12.* **Test accuracy vs. total training epochs for ResNet20×{1}/CIFAR-100 with SET**. Our method consistently outperforms the baseline across all sparsity levels, achieving higher generalization accuracy and demonstrating improved rate of convergence.

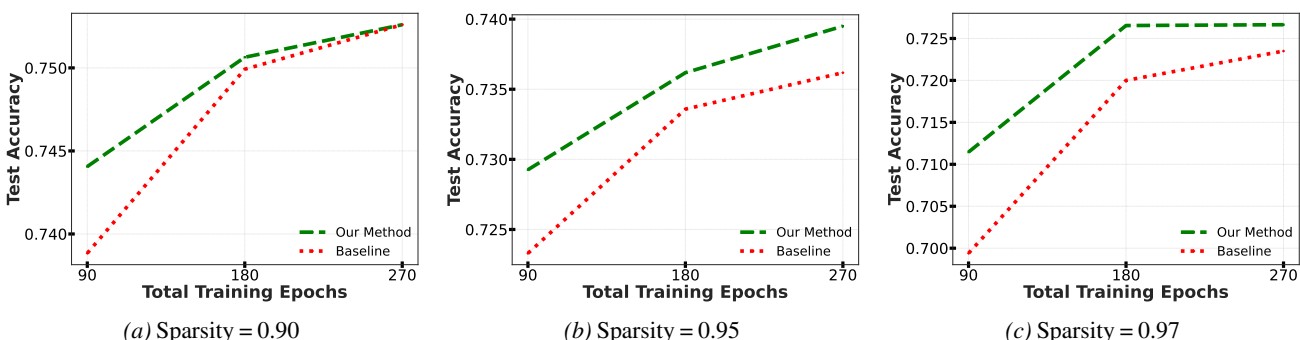

*Figure 13.* **Test accuracy (top-1) vs. total training epochs for ResNet50/ImageNet with RigL**. Our method consistently outperforms the baseline across all sparsity levels, achieving higher generalization accuracy and demonstrating improved rate of convergence.

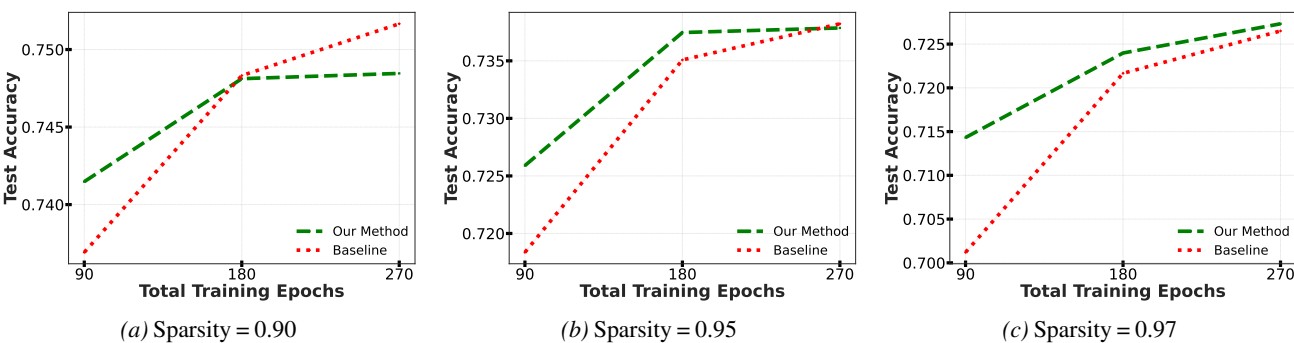

*Figure 14.* **Test accuracy (top-1) vs. total training epochs for ResNet50/ImageNet with SET**. Our method mostly outperforms the baseline across all sparsity levels, achieving higher generalization accuracy and demonstrating improved rate of convergence.

## C.3. Training Curves

In this section, we present additional training and test accuracy trajectories for ImageNet experiments using SET . These curves illustrate the detailed convergence dynamics across the entire training duration for varying sparsity levels, as shown in Figure 15 and Figure 16.

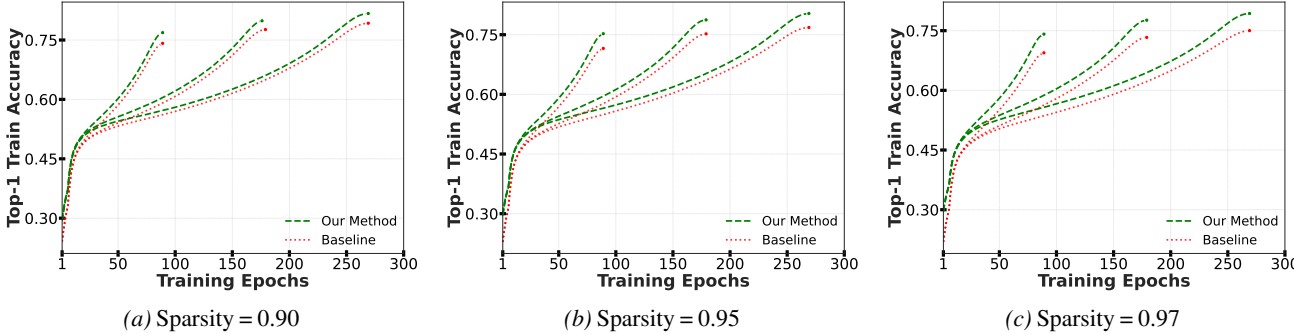

*(a)* Sparsity = 0.90        *(b)* Sparsity = 0.95        *(c)* Sparsity = 0.97

*Figure 15.* **Train accuracy (top-1) vs. epochs on ImageNet with SET**. As observed, our method significantly improves the training dynamics and convergence of SET, especially for higher sparsities.

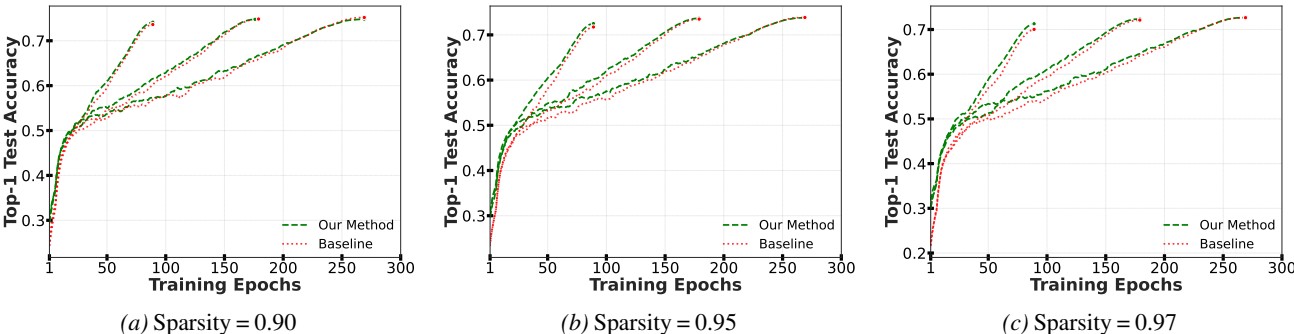

*(a)* Sparsity = 0.90        *(b)* Sparsity = 0.95        *(c)* Sparsity = 0.97

*Figure 16.* **Test accuracy (top-1) vs. epochs on ImageNet with SET**. Our method converges faster and achieves higher accuracy with fewer training epochs, particularly at higher sparsity levels. With much longer training schedules, both methods converge to similar final accuracy.

## C.4. Experiments with HAM Optimizer

In this section, we empirically validate that our proposed sparsity-aware gradient scaling is complementary to the Hyperbolic Aware Minimization (HAM) optimizer. Experimental results on CIFAR-100, using both RigL and SET, demonstrate that integrating our method with HAM further accelerates convergence and enhances generalization performance across all sparsities, as shown in Figure 17 and Figure 18. **Note**: The x-axis represents different models trained independently for different number of total number of epochs.

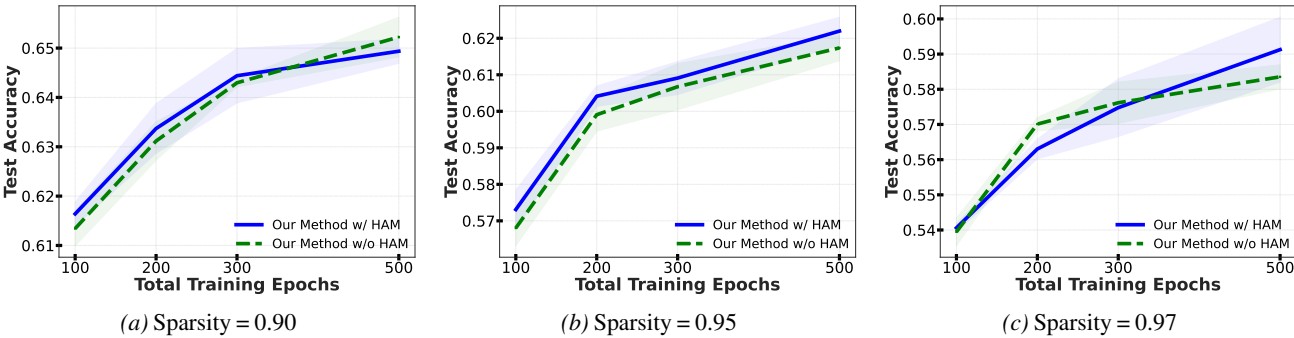

*(a)* Sparsity = 0.90      *(b)* Sparsity = 0.95      *(c)* Sparsity = 0.97

*Figure 17.* **Test accuracy vs. total training epochs for ResNet20×{1}/CIFAR-100 with RigL**. We evaluate the compatibility of our method w/ HAM optimization demonstrating improved rate of convergence across increasing sparsity levels.

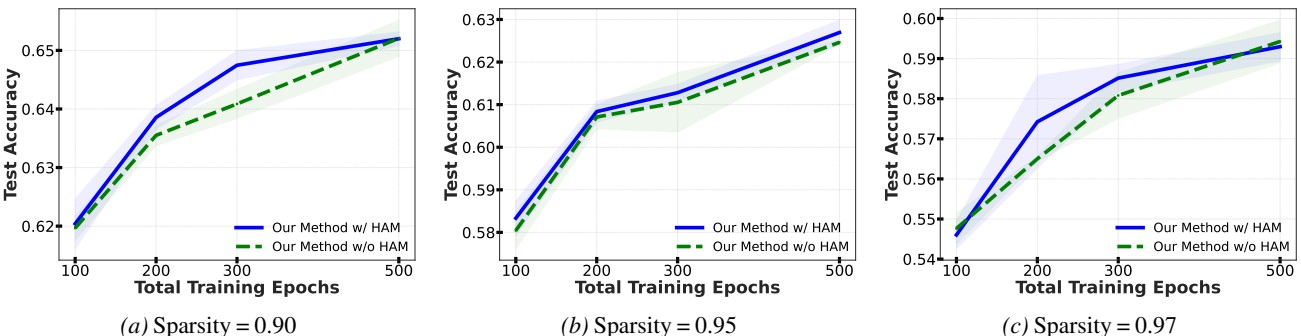

*(a)* Sparsity = 0.90      *(b)* Sparsity = 0.95      *(c)* Sparsity = 0.97

*Figure 18.* **Test accuracy vs. total training epochs for ResNet20×{1}/CIFAR-100 using SET**. We evaluate the compatibility of our method w/ HAM optimization demonstrating improved rate of convergence across increasing sparsity levels.

## D. Derivation for LN with Uniform Sparsity

**Setup:** We analyze the gradient flow for LN under the assumption of *uniform sparsity*, where every neuron $i$ has the same sparsity level $s$ (i.e., $s_i = s \; \forall i$). Let $x_i$ denote the unnormalized feature and $x_i'$ denote the feature after masking. Following a similar analysis for LN, we derive the gradients for each neuron as follows:

$$\frac{\partial L}{\partial \gamma} = \sum_i \frac{\partial L}{\partial y_i} \frac{x_i' - \mu}{\sqrt{\sigma_{LN}^2 + \epsilon}}$$

$$\frac{\partial L}{\partial \beta} = \sum_i \frac{\partial L}{\partial y_i}$$

$$\frac{\partial L}{\partial x_i'} = \sum_j \frac{\partial L}{\partial y_j} \left[ \gamma \left( \frac{\mathcal{N}_l \delta_{ij} - 1}{\mathcal{N}_l \sigma_{LN}} - \frac{(x_i' - \mu)(x_j' - \mu)}{\sigma_{LN}^3} \right) \right],$$

where $L$ is the loss function, $y_i$ is the $i^{\text{th}}$ normalized feature, $\delta_{ij}$ is the Kronecker delta, $\mathcal{N}_l$ is the number of neurons in layer $l$, and and $\sigma_{LN}$ is the layer's standard deviation.. For each iteration, assuming the features are i.i.d. distributed, i.e., $\mathbf{cov}(x_i', x_j') = 0 \; \forall \, i \neq j$, gradients averaged over a batch can be written as:

$$\mathbb{E}\left[ \frac{\partial L}{\partial x_i'} \right] = \mathbb{E}\frac{\partial L}{\partial y_i} \left[ \gamma \left( \frac{\mathcal{N}_l - 1}{\mathcal{N}_l \sigma_{LN}} - \frac{(x_i' - \mu)^2}{\sigma_{LN}^3} \right) \right] + \mathbb{E}\sum_{j \neq i} \frac{\partial L}{\partial y_j} \left[ \gamma \left( \frac{-1}{\mathcal{N}_l \sigma_{LN}} - \frac{(x_i' - \mu)(x_j' - \mu)}{\sigma_{LN}^3} \right) \right]$$

$$= \frac{\partial L}{\partial y_i} \left[ \gamma \left( \frac{\mathcal{N}_l - 1}{\mathcal{N}_l \sigma_{LN}} - \frac{\text{Var}(x_i')}{\sigma_{LN}^3} \right) \right] + \sum_{j \neq i} \frac{\partial L}{\partial y_j} \left[ \gamma \left( \frac{-1}{\mathcal{N}_l \sigma_{LN}} \right) \right] \tag{14}$$

**Impact of Uniform Sparsity:** Consequently, the number of incoming connections is constant across the layer. From Equation (7) in Section 3, the pre-activation variance depends on the sparsity and the fan-in from the previous layer. Since $s$ is constant, the variance is identical for every sparse neuron $i$:

$$\text{Var}(x_i') = (1 - s)\text{Var}(x_i), \tag{15}$$

where $\text{Var}(x_i) = \sigma_i^2$ corresponds to the variance of the fully-connected neuron. We compute the expected variance of the LN statistics, $\sigma_{LN}^2$. Since the individual neuron variances are identical, the summation over the feature dimension simplifies to:

$$\sigma_{LN}^2 \approx \frac{1}{\mathcal{N}_l} \sum_{k=1}^{\mathcal{N}_l} \text{Var}(x_k') = \frac{1}{\mathcal{N}_l} \sum_{k=1}^{\mathcal{N}_l} (1 - s)\text{Var}(x_i) = \frac{1}{\mathcal{N}_l} (1 - s)\text{Var}(x_i) \sum_{k=1}^{\mathcal{N}_l} 1 = (1 - s)\sigma_i^2. \tag{16}$$

Plugging in $\text{Var}(x_i')$ and $\sigma_{LN}$ from Equations (15) and (16) into Equation (14), sparse gradients with LN can be derived. Note that the ratio $\frac{\text{Var}(x_i')}{\sigma_{LN}^2} = \frac{(1-s)\text{Var}(x_i)}{(1-s)\sigma_i^2}$ remains invariant, simplifying the bracketed term:

$$\mathbb{E}_s\left[ \frac{\partial L}{\partial x_i'} \right] = \frac{\partial L}{\partial y_i} \left[ \gamma \left( \frac{\mathcal{N}_l - 1}{\mathcal{N}_l \sigma_{LN}} - \frac{\text{Var}(x_i')}{\sigma_{LN}^3} \right) \right] + \sum_{j \neq i} \frac{\partial L}{\partial y_j} \left[ \gamma \left( \frac{-1}{\mathcal{N}_l \sigma_{LN}} \right) \right]$$

$$= \frac{\partial L}{\partial y_i} \frac{\gamma}{\sigma_{LN}} \left[ \frac{\mathcal{N}_l - 1}{\mathcal{N}_l} - \frac{\text{Var}(x_i')}{\sigma_{LN}^2} \right] + \sum_{j \neq i} \frac{\partial L}{\partial y_j} \frac{\gamma}{\sigma_{LN}} \left[ \frac{-1}{\mathcal{N}_l} \right]$$

$$= \frac{\partial L}{\partial y_i} \frac{\gamma}{\sqrt{(1-s)\sigma_i^2}} \underbrace{\left[ \frac{\mathcal{N}_l - 1}{\mathcal{N}_l} - \frac{(1-s)\text{Var}(x_i)}{(1-s)\sigma_i^2} \right]}_{\text{Invariant Term}} + \sum_{j \neq i} \frac{\partial L}{\partial y_j} \frac{\gamma}{\sqrt{(1-s)\sigma_i^2}} \left[ \frac{-1}{\mathcal{N}_l} \right]$$

$$= \frac{1}{\sqrt{1-s}} \left( \mathbb{E}_{dense}\left[ \frac{\partial L}{\partial x_i} \right] \right)$$

Therefore, under uniform sparsity, the sparse gradient $\mathbb{E}_s\left[ \frac{\partial L}{\partial x_i'} \right]$ is scaled uniformly by a factor of $\frac{1}{\sqrt{(1-s)}}$. $\square$

# E. Experimental Details

In this section, we provide detailed hyperparameter settings, which were determined via grid search for both the baselines and the proposed method to ensure reproducibility.

## E.1. Hyperparamters

*Table 4.* Hyperparameters for ResNet20 on CIFAR-100 experiments (RigL/SET).

| Hyperparameter | Value |
|---|---|
| Optimizer | SGD |
| Batch Size ($B$) | 256 |
| Learning Rate ($\eta$) | 0.1 |
| Learning Rate Schedule | Cosine Decay |
| Momentum | 0.9 |
| Weight Decay | $5 \times 10^{-4}$ |
| *Sparsity Parameters* | |
| Sparsity Distribution | ERK |
| Drop Fraction ($f_{\text{drop}}$) | 0.3 |
| Update Frequency ($\Delta T$) | 100 iterations |
| HAM Hyperparameter ($h_1$) | 80 |

*Table 5.* Hyperparameters for ResNet50 on ImageNet experiments (RigL/SET).

| Hyperparameter | Value |
|---|---|
| Optimizer | SGD |
| Batch Size ($B$) | 128 (scaled by factor) |
| Base Learning Rate ($\eta$) | 0.1 |
| Momentum | 0.9 |
| Weight Decay | $1 \times 10^{-4}$ |
| *Sparsity Parameters* | |
| Sparsity Distribution | ERK |
| Drop Fraction ($f_{\text{drop}}$) | 0.3 |
| Update Frequency ($\Delta T$) | 100 iterations |

## E.2. Learning Rate Schedules

### E.2.1. IMAGENET

For ImageNet experiments, we use a **cosine decay schedule with a linear warmup**, implemented via `optax.join_schedules`. The schedule consists of two distinct phases:

**1. Peak Learning Rate Scaling.** The peak learning rate ($\eta_{peak}$) is dynamically scaled based on the global batch size ($B$) to ensure consistent convergence across different hardware configurations. It is calculated relative to a base batch size of 256:

$$\eta_{peak} = \eta_{base} \times \frac{B}{256},$$

where $\eta_{base} = 0.1$ is the base learning rate provided in the arguments.

**2. Warmup Phase.** Training begins with a linear warmup phase lasting for 5 **epochs** ($T_{warmup}$). During this period, the learning rate increases linearly from a small initial value ($\eta_{init}$) to the peak learning rate:

$$\eta_t = \text{Linear}(\eta_{init}, \eta_{peak}, t) \quad \text{for } 0 \leq t < T_{warmup},$$

where $\eta_{init} = 1 \times 10^{-5}$.

**3. Cosine Decay Phase.** After the warmup phase, the learning rate follows a standard cosine decay schedule for the remainder of the training duration ($T_{total} - T_{warmup}$). The learning rate decays from $\eta_{peak}$ down to a final minimum value ($\eta_{end}$) by the last epoch:

$$\eta_t = \eta_{end} + \frac{1}{2}(\eta_{peak} - \eta_{end})\left(1 + \cos\left(\frac{t - T_{warmup}}{T_{total} - T_{warmup}}\pi\right)\right),$$

where $\eta_{end} = 1 \times 10^{-5}$ and $T_{total}$ is the total number of training epochs (e.g., 90, 180, or 270).

### E.2.2. CIFAR-100

For CIFAR-100 experiments, we use a **cosine decay schedule with a linear warmup**. Unlike the ImageNet schedule, the base learning rate is determined directly by the hyperparameters without dynamic batch size scaling. The schedule proceeds in two phases:

**1. Warmup Phase.** Training initiates with a linear warmup phase for the first 5 **epochs** ($T_{warmup}$). The learning rate increases linearly from 0 to the base learning rate ($\eta_{base}$):

$$\eta_t = \eta_{base} \times \frac{t}{T_{warmup}} \quad \text{for } 0 \leq t < T_{warmup},$$

where $\eta_{base}$ is the learning rate provided in the arguments (typically 0.1).

**2. Cosine Decay Phase.** Following the warmup, the learning rate follows a standard cosine annealing schedule for the remaining epochs ($T_{total} - T_{warmup}$). The learning rate decays from $\eta_{base}$ to a final minimum value ($\eta_{end}$):

$$\eta_t = \eta_{end} + \frac{1}{2}(\eta_{base} - \eta_{end})\left(1 + \cos\left(\frac{t - T_{warmup}}{T_{total} - T_{warmup}}\pi\right)\right),$$

where $\eta_{end} = 1 \times 10^{-6}$ and $T_{total}$ is the total number of training epochs (e.g., 100, 200, 300, or 500).

### E.3. Empirical Analysis of BN

In Figure 2, we use a two-layer MLP on MNIST with a masked dense input layer (64 hidden units), optional BatchNorm, ReLU, and a 10-way output layer. Images are normalized with mean 0.1307 and std 0.3081 and flattened to 784-D vectors. For each sparsity level $s \in \{0, 0.1, ... 0.95\}$, we instantiate a dense model and a sparse model that share the same random initialization; the sparse model applies a random mask at initialization. We compute gradients of the first layer's kernel on 100 mini-batches (batch size 64) at random initialization (no training or optimizer updates). The empirical ratio is computed as the mean absolute sparse gradient divided by the mean absolute dense gradient restricted to the active (non-zero) entries. We repeat the procedure in two settings—one with BatchNorm after the first layer and one without BatchNorm—to isolate how BatchNorm affects gradient magnitudes in sparse models.

## F. Ablation: Effect of Gradient Direction

Our proposed method rescales gradients by $\sqrt{1-s_i}$, which modifies both the vector's direction and its norm. To isolate the directional component, we use a **gradient renormalization** mechanism—implemented directly in the optimization loop. This mechanism computes the global $\ell_2$ norm $\|\mathbf{g}\|_2$ over all parameter gradients and applies a scalar multiplier of $1/\max(\|\mathbf{g}\|_2, 1)$, effectively constraining the global gradient norm to a maximum of $1$ (in PyTorch, this is commonly referred to as "gradient clipping"). This ensures that the optimizer takes steps of comparable magnitude to the baseline, forcing any performance gains to stem solely from the corrected descent direction.

### F.1. Comparison of Test Accuracy vs. Total Training Epochs Plots w/ Gradient Renormalization

In this section, we evaluate the generalization performance of our method w/ gradient renormalization against the baseline by training independent models across varying total training epochs. We demonstrate that our method consistently achieves better convergence efficiency and final accuracy across all evaluated sparsity levels, as shown in Figure 19, Figure 20, Figure 21, and Figure 22. **Note**: The x-axis represents different models trained independently for different number of total number of epochs.

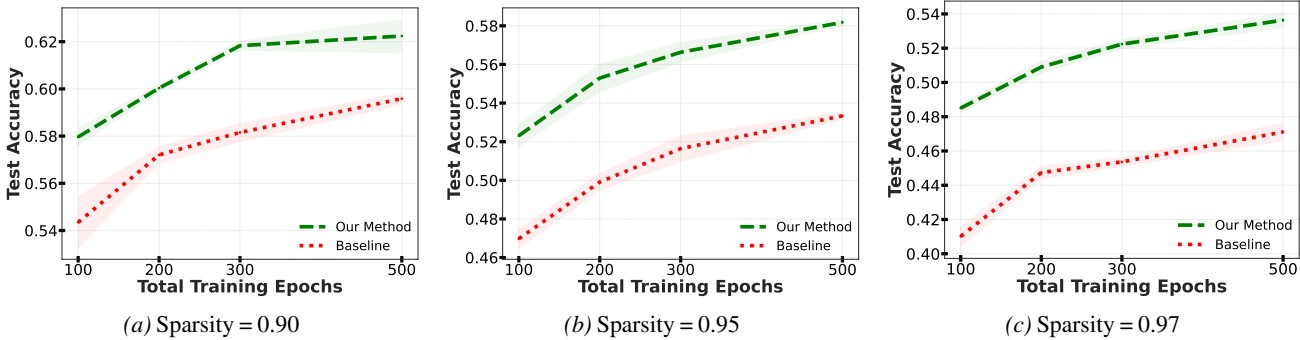

*(a)* Sparsity = 0.90      *(b)* Sparsity = 0.95      *(c)* Sparsity = 0.97

*Figure 19.* **Test accuracy vs. total training epochs for ResNet20×{1}/CIFAR-100 with RigL**. We compare our proposed sparsity-aware gradient scaling method against the standard RigL baseline w/ gradient renormalization across increasing sparsity levels.

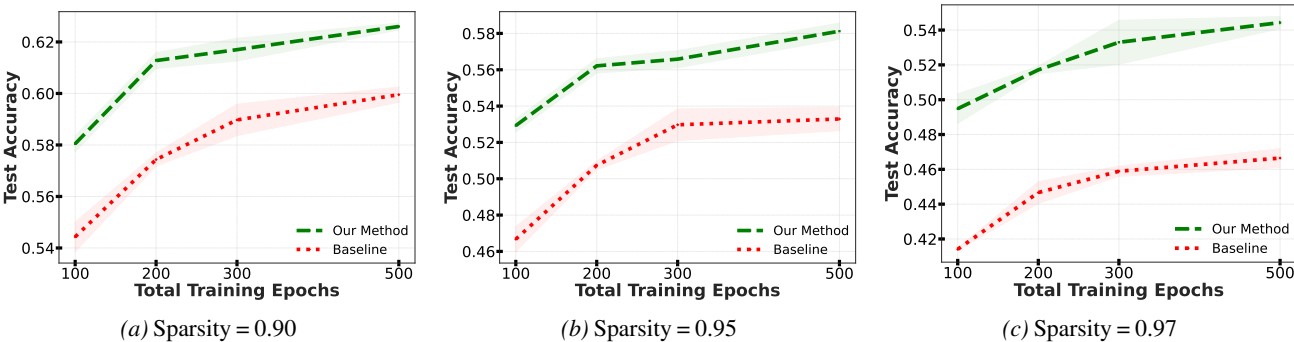

*(a)* Sparsity = 0.90      *(b)* Sparsity = 0.95      *(c)* Sparsity = 0.97

*Figure 20.* **Test accuracy vs. total training epochs for ResNet20×{1}/CIFAR-100 with SET**. We compare our proposed sparsity-aware gradient scaling method against the standard SET baseline w/ gradient renormalization across increasing sparsity levels.

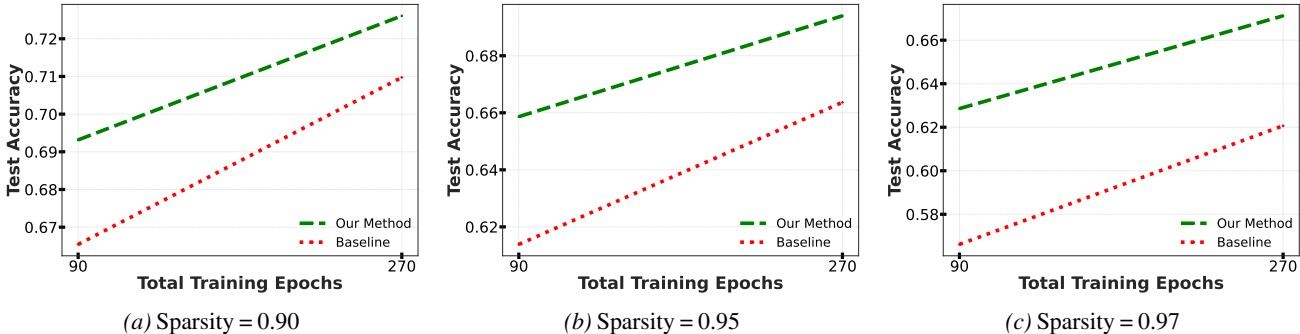

*(a)* Sparsity = 0.90      *(b)* Sparsity = 0.95      *(c)* Sparsity = 0.97

*Figure 21.* **Test accuracy (top-1) vs. total training epochs for ResNet50/ImageNet with RigL**. We compare our proposed sparsity-aware gradient scaling method against the standard RigL baseline w/ gradient renormalization across increasing sparsity levels to only analyze the effect of gradient direction. As shown training with our method improves convergence rate, i.e. models achieve better generalization with less training epochs.

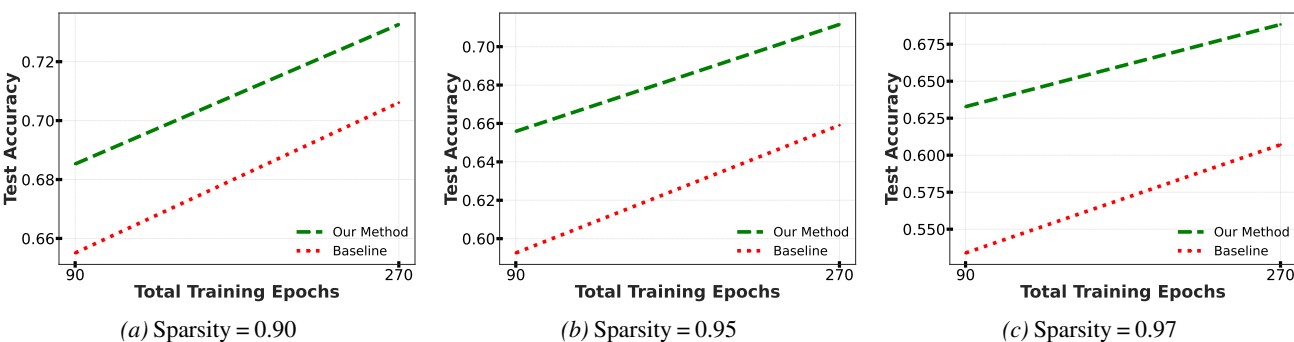

*(a)* Sparsity = 0.90      *(b)* Sparsity = 0.95      *(c)* Sparsity = 0.97

*Figure 22.* **Test accuracy (top-1) vs. total training epochs for ResNet50/ImageNet with SET**. We compare our proposed sparsity-aware gradient scaling method against the standard SET baseline w/ gradient renormalization across increasing sparsity levels to only analyze the effect of gradient direction. As shown training with our method improves convergence rate, i.e. models achieve better generalization with less training epochs.

## F.2. Results w/ Gradient Renormalization

We present a detailed numerical comparison of test accuracy across various sparsity levels for both CIFAR-100 and ImageNet. These results demonstrate the consistent performance improvements of our proposed method w/ gradient renormalization over standard baselines, as shown in Table 6, Table 7, Table 8, and Table 9.

*Table 6.* **Test accuracy (top-1) on ImageNet with RigL**. Our method consistently achieves better generalization than the baseline, especially with a smaller training budget.

| Sparsity | Method | Training Epochs | |
|---|---|---|---|
| | | 90 | 270 |
| 90% | Ours | **69.32** | **72.61** |
| | Baseline | 66.54 | 70.97 |
| 95% | Ours | **65.86** | **69.40** |
| | Baseline | 61.39 | 66.37 |
| 97% | Ours | **62.86** | **67.12** |
| | Baseline | 56.62 | 62.07 |

*Table 7.* **Test accuracy (top-1) on ImageNet with SET**. Our method consistently achieves better generalization than the baseline, especially with a smaller training budget.

| Sparsity | Method | Training Epochs | |
|---|---|---|---|
| | | 90 | 270 |
| 90% | Ours | **68.53** | **73.26** |
| | Baseline | 65.51 | 70.61 |
| 95% | Ours | **65.59** | **71.16** |
| | Baseline | 59.25 | 65.91 |
| 97% | Ours | **63.28** | **68.83** |
| | Baseline | 53.39 | 60.71 |

*Table 8.* **Test accuracy on CIFAR-100 dataset with RigL**. Our proposed method improves the convergence across different sparsity levels and achieves higher generalization; the baseline takes many more training epochs to converge to similar accuracy.

| Sparsity | Method | Training Epochs | | | |
|---|---|---|---|---|---|
| | | 100 | 200 | 300 | 500 |
| 90% | Ours (w/ HAM) | **59.57±0.50** | **61.81±0.23** | **62.98±0.04** | **63.87±0.24** |
| | Ours (w/o HAM) | 57.96±0.37 | 60.03±0.14 | 61.83±0.19 | 62.24±0.71 |
| | Baseline (w/ HAM) | 56.97±0.40 | 59.30±0.37 | 60.93±0.11 | 62.08±0.55 |
| | Baseline | 54.35±1.08 | 57.20±0.34 | 58.15±0.39 | 59.59±0.24 |
| 95% | Ours (w/ HAM) | **54.25±0.28** | **57.61±0.27** | **58.51±0.19** | **59.83±0.33** |
| | Ours (w/o HAM) | 52.31±0.56 | 55.29±0.70 | 56.63±0.52 | 58.18±0.16 |
| | Baseline (w/ HAM) | 50.83±0.35 | 53.26±0.21 | 54.20±1.23 | 55.85±0.26 |
| | Baseline | 46.97±0.47 | 49.91±0.44 | 51.64±0.66 | 53.34±0.17 |
| 97% | Ours (w/ HAM) | **50.65±0.46** | **52.82±0.41** | **54.18±0.44** | **54.93±0.25** |
| | Ours (w/o HAM) | 48.50±0.23 | 50.90±0.35 | 52.24±0.27 | 53.64±0.43 |
| | Baseline (w/ HAM) | 45.29±0.49 | 48.16±0.19 | 49.37±0.49 | 50.06±0.26 |
| | Baseline | 41.01±0.60 | 44.74±0.36 | 45.36±0.25 | 47.11±0.50 |

*Table 9.* **Test accuracy on CIFAR-100 dataset with SET**. Our proposed method improves the convergence across different sparsity levels and achieves higher generalization; the baseline takes many more training epochs to converge to similar accuracy.

| Sparsity | Method | Training Epochs | | | |
|---|---|---|---|---|---|
| | | 100 | 200 | 300 | 500 |
| 90% | Ours (w/ HAM) | **59.22±0.46** | **62.05±0.37** | **62.93±0.11** | **63.39±0.50** |
| | Ours (w/o HAM) | 58.05±0.31 | 61.28±0.30 | 61.70±0.44 | 62.60±0.13 |
| | Baseline (w/ HAM) | 57.41±0.43 | 59.61±0.46 | 60.53±0.31 | 61.10±0.21 |
| | Baseline | 54.43±0.57 | 57.43±0.25 | 58.97±0.61 | 59.95±0.30 |
| 95% | Ours (w/ HAM) | **54.66±0.75** | **57.50±0.54** | **59.01±0.59** | **59.73±0.17** |
| | Ours (w/o HAM) | 52.93±0.35 | 56.21±0.39 | 56.58±0.47 | 58.12±0.45 |
| | Baseline (w/ HAM) | 50.11±0.53 | 53.06±0.38 | 54.79±0.28 | 55.96±0.51 |
| | Baseline | 46.67±0.69 | 50.74±0.23 | 52.97±0.87 | 53.29±0.64 |
| 97% | Ours (w/ HAM) | **50.54±0.45** | **53.47±0.29** | **54.07±0.12** | **55.84±0.36** |
| | Ours (w/o HAM) | 49.48±0.84 | 51.71±0.22 | 53.30±1.26 | 54.43±0.35 |
| | Baseline (w/ HAM) | 45.14±0.30 | 48.14±0.19 | 48.83±0.77 | 49.51±0.72 |
| | Baseline | 41.41±0.07 | 44.67±0.62 | 45.89±0.30 | 46.65±0.54 |

## F.3. Training Curves w/ Gradient Renormalization

In this section, we present additional training and test accuracy trajectories for ImageNet experiments using RigL. These curves illustrate the detailed convergence dynamics across the entire training duration for varying sparsity levels, as shown in Figure 23 and Figure 24.

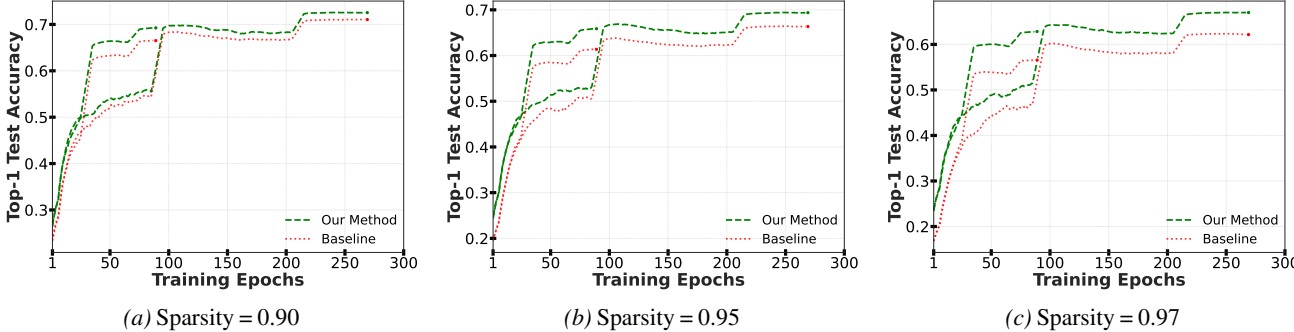

*(a)* Sparsity = 0.90          *(b)* Sparsity = 0.95          *(c)* Sparsity = 0.97

*Figure 23.* **Test accuracy (top-1) vs. epochs on ImageNet with RigL**. Our method converges faster and achieves higher accuracy with fewer training epochs, particularly at higher sparsity levels.

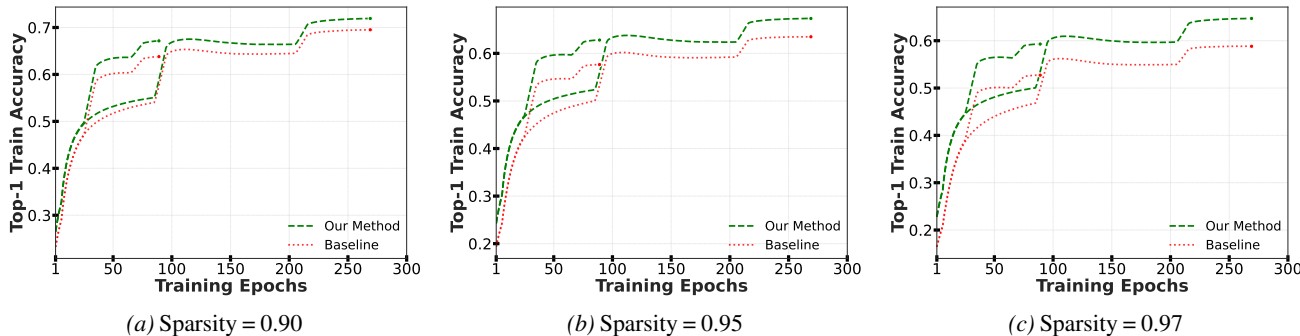

*(a)* Sparsity = 0.90          *(b)* Sparsity = 0.95          *(c)* Sparsity = 0.97

*Figure 24.* **Train accuracy (top-1) vs. epochs on ImageNet with RigL**. As observed, our method significantly improves the training dynamics and convergence of RigL, especially for higher sparsities.

## F.4. Experiments with HAM Optimizer w/ Gradient Renormalization

In this section, we empirically validate that our proposed sparsity-aware gradient scaling w/ gradient renormalization is complementary to the Hyperbolic Aware Minimization (HAM) optimizer. Experimental results on CIFAR-100, using both RigL and SET, demonstrate that integrating our method with HAM further accelerates convergence and enhances generalization performance across all sparsities, as shown in Figure 25 and Figure 26. **Note**: The x-axis represents different models trained independently for different number of total number of epochs.

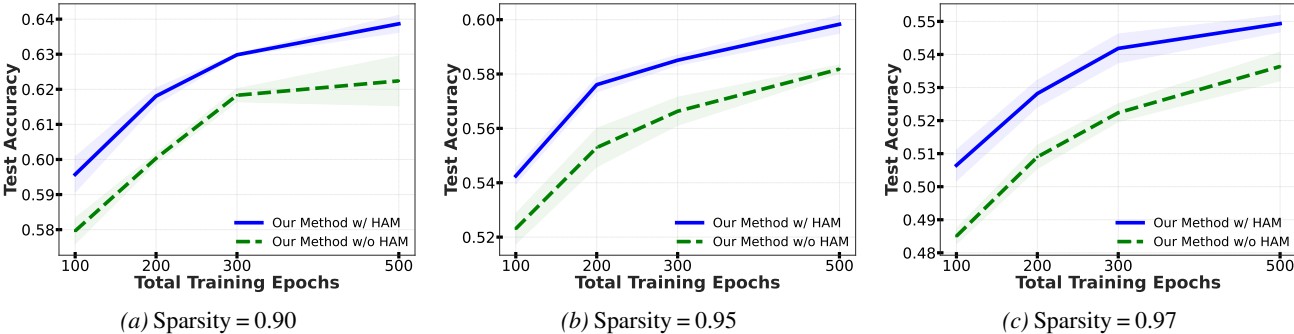

*(a)* Sparsity = 0.90      *(b)* Sparsity = 0.95      *(c)* Sparsity = 0.97

*Figure 25.* **Test accuracy vs. total training epochs for ResNet20×{1}/CIFAR-100 with RigL**. We evaluate compatibility of our method w/ HAM optimization demonstrating improved rate of convergence across increasing sparsity levels.

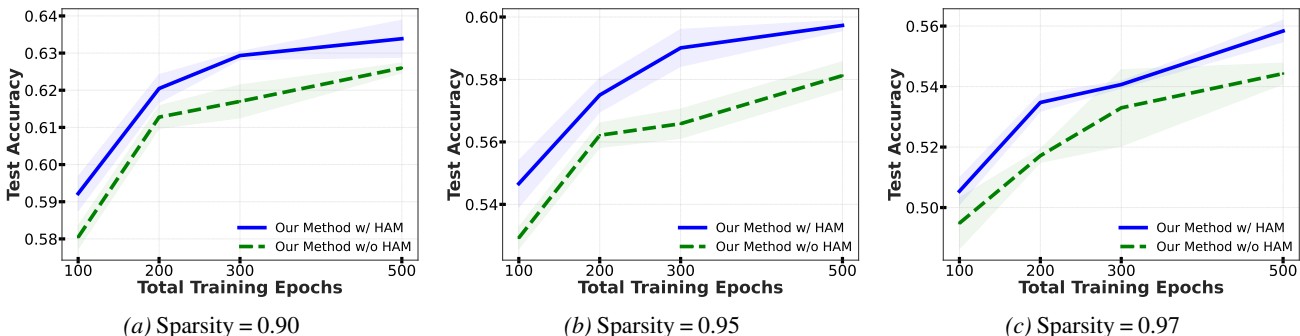

*(a)* Sparsity = 0.90      *(b)* Sparsity = 0.95      *(c)* Sparsity = 0.97

*Figure 26.* **Test accuracy vs. total training epochs for ResNet20×{1}/CIFAR-100 with SET**. We evaluate compatibility of our method w/ HAM optimization demonstrating improved rate of convergence across increasing sparsity levels.

