# OpenReview forum: "SparseOpt: Addressing Normalization-induced Gradient Skew in Sparse Training"
_ICML.cc/2026/Conference — ICML 2026 regular_

### Official Review · Reviewer_8Y4v · 2026-03-05

**Soundness:** 4
**Presentation:** 3
**Significance:** 3
**Originality:** 3
**Overall Recommendation:** 5
**Confidence:** 4

**Summary:**

This paper proposes that Batch Normalization adversely affects sparse training and corrects this sparse training with The authors propose SparseOpt, a diagonal preconditioner that corrects this skew, and demonstrate consistently faster convergence on ResNet models across CIFAR-100 and ImageNet.

This is a well-done paper. It identifies a real and previously overlooked problem — BN's non-uniform gradient scaling under sparsity — and provides a clean, minimal fix with consistent empirical gains. The analysis is clear, the experiments are thorough, and the method composes well with existing approaches.

**Compliance With Llm Reviewing Policy:**

Affirmed.

**Key Questions For Authors:**

The ITOP analysis shows that original gradients are better for regrowth but corrected gradients are better for weight updates. Does this suggest that BN's gradient distortion is actually *helpful* for mask exploration? If so, is there a principled way to exploit this rather than treating it as a nuisance?

Under what conditions might the analytical scaling (Eq. 11) deviate from the actual gradient distortion in practice — e.g., with correlated weights, non-i.i.d. activations, or structured sparsity patterns?

**Limitations:**

**How is the per-neuron sparsity s_i estimated in practice?** It is presumably computed directly from the binary mask, which makes it exact rather than estimated. But the paper does not discuss this explicitly. For settings where the mask changes frequently or where structured sparsity patterns create correlations, the relationship between the mask-derived s_i and the actual variance reduction may not hold exactly. Some discussion of when the analytical scaling (Eq. 11) might deviate from practice would be valuable.

**Strengths And Weaknesses:**

The basic is proven correctly, the derivation (Eq. 10-11) is straightforward and the empirical validation in Figure 2 confirms the analytical prediction exactly, a good contribution.

The skewing is given by an inverse of the variance, the fix is to multiply by a relative factor to fix. it, simple and elegant.

Multiple DST methods (RigL, SET), multiple datasets (CIFAR-100, ImageNet), multiple sparsity levels, multiple training schedules, and comparison with SOTA (HAM). The improvement is consistent. Showing that corrected gradients for weight updates but original gradients for mask regrowth works best reveals a subtle interplay between optimization and topology search is appreciated.

---

> ### Author Rebuttal · Authors · 2026-03-28
>
> Thank you for the feedback, we are glad that you found our paper interesting and found our method elegant. We provide further details to answer your concerns and questions:
>
> * **Does this suggest that BN's gradient distortion is actually helpful for mask exploration?**
>
> We also found this observation interesting but not surprising. RigL uses gradient norm to regrow/reactivate during the mask update step. Liu et al. showed that mask exploration helps dynamic sparse training [1].
>
> BN increases the grad norm of sparser neurons, which implies weights/params of sparser neurons will have higher chances of getting reactivated again (for RigL). This explains why original gradients explore more, as they encourage sparse neurons to get reactivated during training, allowing DST to explore a larger set of sparse topologies/ITOP rate. This also explains why original gradients help mask exploration, while corrected gradients for training/weight update steps work best in practice. While corrected gradients help in optimization, as seen from both our analytical analysis and empirical results, skewed gradients implicitly help in mask exploration.
>
> Note: SET reactivates params randomly, not based on grad norm; our proposed method still improves convergence speed. We intentionally added SET in our results to disentangle the effect of grad on optimization and mask exploration. Experiments on SET also show consistent improvement over baseline with our proposed optimizer, demonstrating that corrected gradients improve convergence speed regardless of mask update strategy (random for SET or grad-based for RigL).
>
> We will add more discussion on this and make this connection more understandable for the readers.
>
> [1] Liu et al., Do We Actually Need Dense Over-Parameterization? In-Time Over-Parameterization in Sparse Training
>
> * **Under what conditions might the analytical scaling (Eq. 11) deviate from the actual gradient distortion in practice**
>
> Our derivation will always hold at the initialization, regardless of architecture, sparsity pattern, as the weights at random initialization are sampled from a normal distribution so the iid condition holds true. While we understand that weights can become more correlated as the training progresses, however, such assumptions are standard in all the initialization papers [1, 2], which propose different initialization methods for improving training dynamics.
> Our empirical results validate this; our proposed method consistently outperforms baselines across different settings.
>
> [1] He et al., Delving Deep into Rectifiers: Surpassing Human-Level Performance on ImageNet Classification
>
> [2] Glorot and Bengio, Understanding the difficulty of training deep feedforward neural networks
>
>
> * **How is the per-neuron sparsity s_i estimated in practice?**
>
> Yes, you’re correct; we compute sparsity from the binary mask, which only needs to be evaluated after the mask update step. We will add details in the paper to make this clearer.
>
> * **For settings where the mask changes frequently or where structured sparsity patterns create correlations, the relationship between the mask-derived s_i and the actual variance reduction may not hold exactly.**
>
> That's a good suggestion.
> We agree that when masks change frequently, or structured sparsity induces correlations, the exact relationship between neuron sparsity and variance reduction may deviate a little. Our analysis is exact at initialization and serves as an approximation during training, consistent with standard assumptions in initialization literature. Our results also show this empirically.
> Importantly, even with such deviations, our insights will still hold --- sparser neurons exhibit lower pre-activation variance, leading BN to amplify their gradients. Our empirical results across settings show that the proposed scaling remains effective in practice. We will clarify this limitation and the discussion in the revision.
>
> We have also added new results and experiments in repsonse to other reviews. We would greatly appreciate it if you could revise your scores if you are satisfied with our answers.

---

> > ### Author Rebuttal · Reviewer_8Y4v · 2026-03-31
> >
> > Thank you for your feedback. Good Luck on your acceptance.

---

> > > ### Author Response · Authors · 2026-04-01
> > >
> > > Please let us know if you any further questions. We would appreciate if you can consider revising the score if you have no further concerns/questions.

---

### Official Review · Reviewer_8KLU · 2026-03-11

**Soundness:** 3
**Presentation:** 2
**Significance:** 2
**Originality:** 3
**Overall Recommendation:** 4
**Confidence:** 3

**Summary:**

This paper studies the interaction between BN and DST. The authors argue that BN introduces gradient scaling that depends on neuron sparsity, which can distort gradient directions during topology updates in sparse networks. To address this issue, the paper proposes SparseOpt, a sparsity-aware preconditioned optimizer that rescales gradients based on neuron sparsity to correct this gradient skew. Experiments on CIFAR-100 and ImageNet with ResNet architectures demonstrate improved convergence speed and slightly better compared to standard SGD in DST settings.

**Compliance With Llm Reviewing Policy:**

Affirmed.

**Final Justification:**

I thank the authors for their detailed response and additional experiments. I appreciate the effort and find that most of my concerns have been addressed. However, I will maintain my original score, as some points still need stronger support.
In particular, the gradient scaling phenomenon has not yet been systematically validated on larger models, and the evidence on static sparse training remains less convincing: the observed improvements appear relatively marginal compared to those in DST, and this difference still lacks sufficient experimental or theoretical support.

**Key Questions For Authors:**

- Can the authors provide empirical measurements of gradient direction changes or optimization noise to support the claim that BN introduces instability in sparse training?

- How sensitive is the proposed method to the learning rate and topology update frequency? Would the method still provide improvements under static sparse training?

- Have the authors evaluated the proposed approach with other normalization layers (e.g., LayerNorm, RMSNorm) or optimizers such as Adam?

**Limitations:**

yes.

**Strengths And Weaknesses:**

### Strengths

- The paper investigates the interaction between BN and sparse training, which is an underexplored but practically important topic. Understanding the optimization behavior of DST methods is valuable for improving the efficiency of sparse neural network training.

- The analysis in Section 3 highlights how heterogeneous connectivity in sparse networks leads to neuron-dependent gradient scaling under Batch Normalization. This provides useful insight into potential optimization instability in sparse training.

- The proposed SparseOpt method is straightforward. The proposed correction can be easily integrated into existing sparse training pipelines.


### Weaknesses

- The paper argues that DST topology updates introduce abrupt changes in neuron sparsity and therefore gradient direction shifts. However, in many DST implementations, regrowth occurs immediately after pruning and gradients are not recomputed between these operations. It is therefore unclear whether the proposed instability mechanism accurately reflects the actual training dynamics.

- The authors claim that BN-induced gradient scaling introduces optimization noise and slows convergence in DST. However, the paper does not provide direct empirical or theoretical evidence demonstrating that this effect significantly harms training stability.

- The gradient scaling phenomenon is mainly illustrated on simplified neural network settings. It is unclear whether the same behavior holds in practical architectures such as ResNet with residual connections and convolutional layers.

- The proposed SparseOpt method resembles sparsity-aware learning rate scaling or gradient normalization techniques used in sparse optimization, like SuPar[1]. The paper does not sufficiently compare SparseOpt with existing methods that adapt learning rates based on sparsity or connectivity.

- The experiments focus only on image classification tasks with SGD-based DST training. It would be useful to evaluate the method under different optimizers (e.g., Adam), different topology update frequencies, or other tasks such as NLP tasks.

[1] Dey N, Bergsma S, Hestness J. Sparse maximal update parameterization: A holistic approach to sparse training dynamics[J]. Advances in Neural Information Processing Systems, 2024, 37: 33836-33862.

---

> ### Author Rebuttal · Authors · 2026-03-28
>
> Thank you for the feedback, we answer your questions below:
>
> * **in DST regrowth occurs after pruning and grad are not recomputed... unclear whether the mechanism accurately reflects the actual training dynamics**
>
> It is correct that grads are not recomputed before regrowth. However, our mechanism arises across iterations due to the mask update, not within the same step. The mask update changes the sparsity pattern and neuron-wise sparsity, and BN rescales gradients in the next backward pass, causing a discontinuous, neuron-dependent change in gradient direction. This effect is independent of the growth strategy: in RigL, updates after regrowth use gradients under the new sparsity; in SET, where growth is random and does not use gradients, our method still improves convergence. We further isolate this by using original grads for mask exploration and corrected grads for weight updates (Sec. 5.2), and still observe faster convergence.
>
> * **the paper does not provide direct empirical or theoretical evidence demonstrating that this effect significantly harms training stability....**
>
> We analytically show the effect of BN on gradients and how it distorts grad. We follow same intuition as many initialization works (Xavier, Glorot, He, etc.) [1,2] that output and gradient norm/variance should remain approx. around 1 and not grow with depth. Our correction similarly ensures grad variance does not explode or shrink as sparsity changes, improving training dynamics. Our results confirm this: convergence speed improves with our correction.
>
> [1] He et al., Delving Deep into Rectifiers: Surpassing Human-Level Performance on ImageNet
> [2] Glorot and Bengio, Understanding the difficulty of training deep feedforward neural nets
>
> * **gradient scaling phenomenon is mainly illustrated on simplified neural network**
>
> Our derivation is architecture-agnostic and makes no structural assumptions. It only assumes that weights are initialized from a normal distribution, as is standard practice. Hence, the same observation will hold for larger models and we will include plots for ResNet in the final version.
>
> * **resembles sparsity-aware learning rate scaling like SuPar**
>
> Thank you for pointing out this work; we will cite it  and discuss more in the related work. While the motivation is similar, our work addresses grad skew caused by norm layers.
> SuPar doesn't account for norm layers or heterogeneous neuron sparsity. Based on μP, it analyzes how to transfer hyperparameters from dense to sparse models and adjusts learning rates based on layer-wise sparsity. It focuses only on static sparse training, where sparsity pattern remains fixed.
> In contrast, we make no such assumption, neuron sparsities are heterogeneous and evolve over time --- enabling improved convergence speed for DST.
>
> * **exps focus only on image tasks with SGD-based DST training... useful to evaluate under different optim (e.g., Adam), different topology update freq, or other task**
>
> In our experiments, we ran a sweep over mask update interval/frequency (and other DST hyperparameters) and observed that the best settings remain the same for both the baseline and our method.
>
> We also compared with optimizers beyond SGD. We chose HAM over Adam, as it is the current SOTA for sparse training and is designed to improve convergence speed. As shown, our method can be combined with HAM and consistently outperforms the baseline; we provide a detailed theoretical analysis in Appendix A1 (page 11). Beyond benchmarking, we also theoretically analyze how our correction interacts with HAM, which could be useful for future sparse training research.
>
> * **Would the method still provide improvements under static sparse training?**
>
> In theory, it should, as the effect of BatchNorm is similar, though less adverse, since neuron sparsity is static. In contrast, in DST, neuron sparsity changes frequently (every 50–100 steps).
> Moreover, in static sparse training, mask choice play a crucial role and to the best of our knowledge, pruning at init/static sparse methods do not work well, as it is theoretically difficult to find the right sparse mask at init [3]. This is why we focus on DST rather than static sparse training.
>
> [3] Kumar et al., No Free Prune: Information-Theoretic Barriers to Pruning
>
> * **Have the authors evaluated the proposed approach with other normalization layers**
>
> We would like to extend this to LayerNorm for language tasks; however, this paper focuses on BatchNorm and vision. In Appendix D, we provide preliminary analysis showing LayerNorm exhibits similar behaviour, i.e, changing grad norms based on sparsity and affecting training dynamics.
> However, correcting this skew is more complex, as LayerNorm normalizes across features, preventing clean factorization for non-uniform sparsity, unlike BN. However, we agree this is important and plan to address it in future work as discussed in sec 6
>
> If you're satasfied with our answers, we kindly ask to revise your final score.

---

> > ### Author Rebuttal · Reviewer_8KLU · 2026-04-01
> >
> > I thank the authors for their response. I still have some concerns that have not been fully addressed.
> >
> > 1. It remains unclear whether the gradient scaling phenomenon also holds for large models.
> >
> > 2. Sup also evaluates on DST scenarios. The paper lacks results on the actual improvement compared with learning rate scaling, making it difficult to assess its effectiveness.
> >
> > 3. There is insufficient evidence on whether the proposed method also works for static sparse training, which falls under the broader claim of sparse training made in the title.

---

> > > ### Author Response · Authors · 2026-04-04
> > >
> > > Thank you for your time and for engaging in the review process. We provide more details and answer your questions below:
> > >
> > > 1. Our derivation will always hold at initialization, where weights are i.i.d. under standard schemes (e.g., Kaiming), making the gradient scaling effect architecture-agnostic, including for conv layers and residual connections. We only used a toy model for simplicity; **all exps are conducted on large-scale models, validating our hypothesis**. We will show gradient scaling illustration for all layers for ResNet in the appendix for the final version to show that our analytical derivation holds for larger models. Here is an example of gradient scaling due to BN in the first conv layer of the third block, which matches the analytic formula: https://postimg.cc/75wyvN5K
> > >
> > > 2. Sparse $\mu$P (SUP) studies how to transfer learning rates across model widths for sparse networks, with the goal of identifying an optimal learning rate. **However, SUP does not adapt the learning rate based on neuron-wise sparsity**, which evolves dynamically during training in DST.  Our work differs from SUP in two key aspects:
> > >
> > > a) **Mechanism and focus**: We address gradient skew induced by normalization layers and propose a correction to address this, which improves convergence speed. In contrast, SUP does not analyze the role of normalization layers; instead, it extends $\mu$P to sparse settings to enable consistent learning rate transfer across widths and sparsities. **For a fair comparison, we perform learning rate sweeps for both baselines and our method, ensuring all results use their respective optimal learning rates**.
> > >
> > > b) **Static vs. dynamic sparsity**: SUP focuses on static sparsity, whereas our work focusses Dynamic Sparse Training (DST), where the mask evolves during training. **This distinction is important, as the interaction between mask updates and normalization layers introduces time-varying gradient distortions that SUP does not take into account**.
> > >
> > > **We also note that the authors of SUP themselves acknowledge this limitation** (Sec. 4.4, page 9,  arXiv version: https://arxiv.org/pdf/2405.15743):
> > >
> > > > “Dynamic sparse methods can make updates to the weight mask such that the distribution of unmasked/non-zero weights changes to something non-Gaussian… Since SµPar assumes weights are drawn from a Gaussian distribution, SµPar ends up ‘overcorrecting’… It would be impactful to develop a parameterization which generalizes SµPar to work for an arbitrary sparse training algorithm.”
> > >
> > > This further highlights that SUP is not designed for DST, whereas our method explicitly addresses this setting. **To provide more evidence, we ran an experiment at high sparsity regime (95%), where sparsity is extremely non-uniform, to empirically compare SUP and SparseOpt**: https://postimg.cc/Lg1x5GbJ
> > >
> > >
> > > 3. We thank the reviewer for this valuable suggestion. In this work, we focus on Dynamic Sparse Training (DST) methods, as they are known to match dense generalization performance, but with significantly slower convergence. In contrast, static sparse training methods, particularly pruning at initialization (PaI), typically do not match dense performance. For this reason, our primary goal is to improve the convergence behavior of DST.
> > >
> > > That said, we agree that evaluating our method on static sparse training would further strengthen the paper. **To this end, we conducted additional experiments with PaI methods (SNIP, SynFlow, and Random) across different sparsity levels**. Our method consistently improves performance, especially in lower sparsity regimes and with masks obtained via SNIP and SynFlow. Preliminary results: https://postimg.cc/8FNmD7Gf
> > >
> > > We note, however, that the underlying challenge differs: since the mask is fixed in PaI, gradient skew is less pronounced, and performance is primarily limited by the quality of the initialization mask rather than optimization dynamics. This further supports our focus on DST, where evolving sparsity induces stronger gradient distortions. We will include these results (including ImageNet experiments) in the appendix for the camera-ready version.
> > >
> > > We appreciate the feedback on the title and will revise both the title and abstract to more clearly emphasize the focus on DST.
> > >
> > > ---
> > > We will carefully incorporate your feedback in the final version. To the best of our knowledge, this is the first work to systematically study the interaction between normalization layers and DST, providing new insights that can guide future research and improve DST as a practical training paradigm.
> > > We hope we have addressed your concerns and would greatly appreciate if you can update your score if you are satisfied with our response.

---

### Official Review · Reviewer_u1ss · 2026-03-12

**Soundness:** 2
**Presentation:** 2
**Significance:** 2
**Originality:** 2
**Overall Recommendation:** 4
**Confidence:** 2

**Summary:**

The authors argue that Batch Normalization, which is usually beneficial in dense networks, behaves differently in sparse networks because different neurons can have different fan-in due to sparsity. The paper proposes SparseOpt, a sparsity-aware preconditioned optimizer. Its main idea is to compensate for the neuron-wise gradient distortion induced by BN by rescaling gradients according to each neuron’s sparsity level, so that updates are better aligned and less disrupted by mask changes.

**Compliance With Llm Reviewing Policy:**

Affirmed.

**Final Justification:**

I appreciate the additional experiments and discussion, and I will increase my score accordingly.

**Key Questions For Authors:**

1. What is the precise novelty of this submission relative to earlier workshop paper, Understanding Normalization Layers for Sparse Training?

2. Can you provide stronger causal evidence that the empirical gains come specifically from correcting BN-induced gradient skew, rather than from a generic gradient-rescaling effect? It would help to see whether similar gains arise from simpler or alternative rescaling baselines.

**Limitations:**

yes

**Strengths And Weaknesses:**

Strengths
The paper is technically coherent at a high level and the proposed optimizer is also well matched to the diagnosis.
The paper is generally well written and easy to follow.

Weaknesses
The paper repeats essentially the same first work claim twice in conclusion, which feels redundant.

It is a targeted contribution for sparse training, especially BN-based DST in vision models.

The claim that this is the first work to highlight the need for sparsity-aware normalization mechanisms seems overstated. The earlier workshop paper Understanding Normalization Layers for Sparse Training [1] already studies normalization in sparse training, including both BN and LN, and introduces a sparsity-aware preconditioned optimizer motivated by that analysis.


[1] Mohammed Adnan, et al. “Understanding Normalization Layers for Sparse Training” ICML 2025 Workshop HiLD.

---

> ### Author Rebuttal · Authors · 2026-03-28
>
> Thank you for the feedback and reviewing our, we answer your questions below:
>
> * **paper repeats essentially the same first work claim twice in conclusion**
>
> Thanks for pointing this out –- we will fix this in the final version of the manuscript.
>
> * **targeted contribution for sparse training, especially BN-based DST in vision models.**
>
> The aim of this work is to highlight that normalization layers adversely affect sparse training dynamics, which have been overlooked by the community (as reviewer **8Y4v**, **8KLU** and **nnYf** pointed out). In this work, we focus on BN, as they are used in vision models, which are still widely used across applications.
>
> * **What is the precise novelty of this submission relative to earlier workshop paper by Adnan et al.**
>
> We will add the workshop paper that you cited in the related work with more discussion. Our method and results differ significantly from the workshop paper you cited. Our derivation in Section 3 is very different, and crucially, the proposed SparseOpt update rule in Eqn. 12 is different; we maintain the gradient norm by dividing by $1/\sqrt{1-s_{avg}}$, which is a completely different mechanism from the workshop paper you cited. Moreover, the workshop paper method only works with models without skip connections. Our method does not make such assumptions, and we validate our method on standard ResNets (w/ skip connections).
>
> We also compare with HAM, the current SOTA sparse training optimizer, and show that our method can be used alongside HAM. We also add a detailed theoretical analysis of how our method/BN interacts with HAM, which we believe will be useful for future research on sparse training optimizers (Appendix A.1). Our empirical results are also more thorough, as noted by reviewer **8Y4v**, "Multiple DST methods (RigL, SET), multiple datasets (CIFAR-100, ImageNet), multiple sparsity levels, multiple training schedules, and comparison with SOTA (HAM). The improvement is consistent."
>
> Furthermore, our paper analyzes the effect of gradient correction on the mask exploration/ITOP rate (Section 5.2), providing new insights into its role in mask exploration. We believe these differences are significant, and the optimizer (the update rule) is different from the cited work, making our contributions and insights novel for the sparse training community.
>
> * **Can you provide stronger causal evidence that the empirical gains come specifically from correcting BN-induced gradient skew, rather than from a generic gradient-rescaling effect?**
>
> Thanks for asking this question. To show that the gain comes directly from correcting the skew or gradient direction/distortion, and not just from gradient rescaling, we ran experiments with gradient clipping/norm (more details Appendix F ). Our experiments clearly show that even with the same gradient norm, when our method only fixes the direction/distortion of the gradient, SparseOpt still significantly improves convergence speed. This strongly suggests that correcting gradient distortion helps sparse training dynamics and establishes a causal link between the two.
>
> We hope we have answered your questions satisfactorily. We have also added new results and experiments in repsonse to other reviews. To the best of our knowledge, our work is the first one to highlight the adverse effect of normalization layers on the sparse training dynamics, which have been overlooked by the sparse training community as others reviewers have also noted. We provide an analytical explanation for this and provide a simple fix, which we believe will be useful for the sparse training community.
>
> We would greatly appreciate it if you could revise your scores if you are satisfied with our answers.

---

> > ### Author Rebuttal · Reviewer_u1ss · 2026-04-03
> >
> > Thank you for the detailed rebuttal. The clarification on the distinction from the workshop paper, especially the difference in the update rule, applicability to ResNets with skip connections, and the additional evidence regarding BN-induced gradient distortion, addresses my main concerns. I appreciate the additional experiments and discussion, and I will increase my score accordingly.

---

> > > ### Author Response · Authors · 2026-04-04
> > >
> > > Thank you for your feedback and updating your score. We will incorporate changes in the final version based on your review.

---

### Official Review · Reviewer_nnYf · 2026-03-13

**Soundness:** 3
**Presentation:** 4
**Significance:** 2
**Originality:** 3
**Overall Recommendation:** 4
**Confidence:** 3

**Summary:**

This paper investigates the slow convergence of Dynamic Sparse Training (DST) and attributes part of the problem to the interaction between Batch Normalization (BN) and heterogeneous neuron sparsity. The authors show that neurons with higher sparsity experience stronger gradient amplification due to BN, which can distort gradient directions during optimization. To address this issue, they propose SparseOpt, a sparsity-aware optimizer that rescales neuron-wise gradients based on neuron density. Experiments on CIFAR-100 and ImageNet with RigL and SET demonstrate improved convergence speed and competitive accuracy, particularly under high sparsity levels.

**Compliance With Llm Reviewing Policy:**

Affirmed.

**Final Justification:**

The rebuttal was well-constructed and clarifies the paper's intent, but because the core issue regarding the practical robustness and magnitude of the theoretical claims has not been completely resolved, I will maintain my current score. I do, however, believe the perspective is interesting and strongly encourage the authors to incorporate the rebuttal discussions into the final manuscript, if accepted.

**Key Questions For Authors:**

- The theoretical analysis relies on simplifying assumptions such as i.i.d. activations and weights, and appears to be validated primarily at initialization. Could the authors clarify whether the proposed gradient scaling behavior persists during later stages of training when these assumptions are likely violated?

- The experimental evaluation is limited to ResNet architectures and mainly considers RigL and SET. Could the authors comment on whether SparseOpt would provide similar benefits for other architectures (e.g., VGG, EfficientNet, MobileNet) or when combined with more recent dynamic sparse training methods?

- The reported improvements on ImageNet appear relatively modest. Could the authors report results across multiple runs or provide standard deviations to better establish statistical significance?

- The paper attributes DST convergence issues to BN-induced gradient skew. However, DST performance is influenced by multiple factors such as topology exploration and mask update dynamics. Could the authors provide additional evidence to isolate the role of gradient skew, for example by measuring gradient statistics during training or evaluating the method under a fixed sparse mask?

- Section 5.2 shows that corrected gradients improve weight updates but degrade mask exploration in RigL. This suggests that the gradient skew identified in the paper may affect optimization and topology exploration differently. Could the authors clarify the role of gradient skew in these two processes and provide further intuition for this trade-off?

**Limitations:**

Please refer to weaknesses

**Strengths And Weaknesses:**

### Strengths:
- The paper provides a theoretical analysis explaining how heterogeneous neuron sparsity interacts with Batch Normalization to induce neuron-dependent gradient scaling.

- The paper introduces an interesting perspective linking the slow convergence of dynamic sparse training (DST) to the interaction between sparsity and BatchNorm, which has been largely overlooked in prior work.

- The proposed SparseOpt method is simple and lightweight, and can be easily integrated with existing DST algorithms such as RigL and SET.

### Weaknesses:
- The theoretical analysis relies on simplifying assumptions such as i.i.d. activations and weights, which may not hold during training as weights develop correlations and activations become non-zero-mean after ReLU. In addition, the theoretical arguments are primarily validated at initialization (Figure 2), leaving it unclear whether the proposed scaling accurately characterizes gradient dynamics throughout optimization.

- The empirical evaluation is limited to ResNet architectures with BatchNorm on image classification benchmarks. It remains unclear whether the proposed observations generalize to other architectures or normalization methods such as LayerNorm, which is discussed in Appendix D but only under a restrictive uniform-sparsity assumption. Additionally, the ImageNet experiments appear to be based on single runs, which limits confidence in the reliability and reproducibility of the reported improvements.

- While the paper hypothesizes that BN-induced gradient skew contributes to the slow convergence of DST, DST optimization is known to be influenced by multiple factors such as topology exploration, gradient noise, and mask update dynamics. The paper does not quantify the relative contribution of gradient skew compared to these other factors, making it difficult to assess how central this issue is to the overall DST convergence problem.

---

> ### Author Rebuttal · Authors · 2026-03-28
>
> Thank you for the feedback, we answer your questions below:
>
> * **theoretical analysis relies on assumptions such as i.i.d. activations** ...
>
> We follow the methodology used in many initialization papers (Xavier init, He init, etc) [1, 2],  which analyzes the grad flow at the random init, where the iid assumption holds true. This still helps in training stability, especially in the early training stage. Our method also makes a similar assumption about the i.i.d. distribution of weights/activation, which holds true at the init. Our empirical results validate our hypothesis across multiple settings.
>
> [1] He et al., Delving Deep into Rectifiers: Surpassing Human-Level Performance on ImageNet Classification
>
> [2] Glorot and Bengio, Understanding the difficulty of training deep feedforward neural nets
>
> * **unclear whether the proposed scaling accurately characterizes grad dynamics throughout optimization**...
>
> We agree strict i.i.d. assumptions weaken as training progresses and features correlate. However, the proposed gradient scaling remains effective. Despite non-zero covariance, pre-activation variance of sparse neurons stays much lower than dense ones, so BN consistently amplifies their grads. Our method acts as a robust correction to this skew, as demonstrated empirically
>
> * **unclear whether the proposed observations generalize to other architectures or normalization methods such as LayerNorm**..
>
> Our analytic observation is architecture-agnostic, as it does not make assumptions about the architecture. We have added additional results on a VGG-style network: https://imgur.com/a/j9Wndbm
>
> We provided preliminary analytical analysis for LayerNorm (Appendix D), which shows that LayerNorm also exhibits the same underlying behavior—it changes grad norms based on sparsity. However, correcting the skew for LayerNorm is more complex, as it normalizes across the feature dim, so we do not obtain a clean factorization of the sparsity-related term for non-uniform sparsity as in the case of BN. LN distorts and scale up grads similar to BN as seen here: https://imgur.com/a/2uxv2AM.
> We agree that addressing LayerNorm is interesting and impactful, and we plan to address this in future work (as discussed in the conclusion sec).
>
> * **ImageNet experiments appear to be based on single runs... provide standard deviations...**
>
> Since we trained across varying schedules for different DST methods and exploration strategies, multiple ImageNet runs were computationally expensive.
> For your ref we report variance for the 90-epoch schedule (SET and RigL) here - https://imgur.com/a/TpyGxVz.
>
> Longer schedules will be included in the camera-ready due to runtime constraints.
>
>
> * **improvements on ImageNet appear relatively modest...**
>
> We would like to highlight that our focus is on improving the convergence speed of DST methods.
> Current DST achieves dense model acc but requires significantly longer training epochs. We show this by comparing acc across diff schedules—demonstrating faster convergence. For longer training, both baseline and our method reach similar acc, as expected since we target convergence speed. Gains at shorter schedules are significant (Table 1); e.g., ImageNet (top-1) improves from **69.9% to 71.1% at 97% sparsity, which is statistically significant**.
>
> * **does not quantify the relative contribution of gradient skew compared to these other factors...**
>
> We agree that DST has multiple sources of training instability (e.g., norm layers, mask update criteria, exploration strategy). Addressing these could further improve convergence speed, as noted in our limitations and left for future work.
> This work specifically focuses on instability due to normalization layers—largely overlooked in sparse training. While normalization helps dense training, it adversely affects sparse training; we believe this is an important contribution and can inform future work in this domain.
>
> *  **corrected gradients improve weight updates but degrade mask exploration in RigL... Could the authors clarify the role of gradient skew in these two processes...**
>
> We find this observation interesting but not surprising. RigL uses grad norm for regrowth, and prior work (ITOP) shows mask exploration aids DST. BN increases grad norms of sparser neurons, making them more likely to be reactivated, thus improving exploration (higher ITOP). This explains why original gradients help mask exploration.
>
> In contrast, corrected gradients improve optimization, as shown in our analysis and experiments—i.e., skewed gradients aid exploration, while corrected gradients improve convergence.
>
> SET uses random regrowth, yet our method still improves convergence, disentangling optimization vs. exploration. Results show consistent gains regardless of mask update strategy.
> We will add more discussion on this and make this connection more understandable for the readers.
>
> We'd greatly appreciate it if you can revise your score if you're satisfied with our response.

---

### Decision · Program_Chairs · 2026-04-30

**Decision:**

Accept (regular)

**Comment:**

This paper (SparseOpt) shows BN (batch norm) skews DST (dynamic sparse training) gradients and improves convergence via a sparsity-aware preconditioner.
Reviewers agree the paper tackles an overlooked DST issue with a simple, useful fix and consistent gains; remaining concerns focus on scope, robustness, and causal isolation.
Main concerns are that the theory relies on assumptions about initializations, limited validation beyond ResNet/BN, ImageNet evidence is not fully multi-run, and the paper only partially isolates BN-induced skew from other DST factors and related sparsity-aware scaling baselines.

Post rebuttal: sufficiently addresses the remaining concerns. I would suggest an acceptance (scores are 5/4/4/4).